# Microfluidics with fluid walls

Edmond J. Walsh[1], Alexander Feuerborn[2], James H.R. Wheeler[3], Ann Na Tan[2], William M. Durham[3,4], Kevin R. Foster [3] & Peter R. Cook [2]

Microfluidics has great potential, but the complexity of fabricating and operating devices has limited its use. Here we describe a method — Freestyle Fluidics — that overcomes many key limitations. In this method, liquids are confined by fluid (not solid) walls. Aqueous circuits with any 2D shape are printed in seconds on plastic or glass Petri dishes; then, interfacial forces pin liquids to substrates, and overlaying an immiscible liquid prevents evaporation. Confining fluid walls are pliant and resilient; they self-heal when liquids are pipetted through them. We drive flow through a wide range of circuits passively by manipulating surface tension and hydrostatic pressure, and actively using external pumps. Finally, we validate the technology with two challenging applications — triggering an inflammatory response in human cells and chemotaxis in bacterial biofilms. This approach provides a powerful and versatile alternative to traditional microfluidics.

[1] Department of Engineering Science, Osney Thermo-Fluids Laboratory, University of Oxford, Osney Mead, Oxford OX2 0ES, UK. [2] The Sir William Dunn School of Pathology, University of Oxford, South Parks Road, Oxford OX1 3RE, UK. [3] Department of Zoology, University of Oxford, South Parks Road, Oxford OX1 3PS, UK. [4] Department of Physics and Astronomy, University of Sheffield, Hounsfield Road, Sheffield S3 7RH, UK. Alexander Feuerborn and James H.R. Wheeler contributed equally to this work. Correspondence and requests for materials should be addressed to E.J.W. (email: edmond.walsh@eng.ox.ac.uk) or to P.R.C. (email: peter.cook@path.ox.ac.uk)

Manipulation of small volumes of liquids is central to many scientific disciplines, including microbiology, cell biology, biochemistry, and materials science. Two popular platforms involve microtiter plates, where liquid is held statically in wells, and microfluidic devices where liquid flows through channels in polydimethylsiloxane (PDMS)[1]. While microtiter plates are widely used, fewer microfluidic devices than expected have been incorporated into scientific workflows[2] despite the demonstrated advantages of the technology[3]. Various reasons are given. Prototyping PDMS-based devices takes at least a few days and is expensive; it also typically requires specialized equipment, a clean room, and advanced training. Once made, devices are usually dedicated to one application, and access to most points in them is limited[2]. Moreover, uncoated PDMS has poor biological and chemical compatibility because it leaches toxins and reacts with organic solvents[2, 4–6]. Air bubbles in conventional devices also present numerous operational challenges: they unbalance flows, damage incorporated cells[7, 8], and trigger molecular aggregation at air-fluid interfaces[9].

Challenges associated with PDMS-based devices are being met using various approaches[2]. For example, methods have been developed to minimize bubble-associated experimental failure[10], while windows in walls[11–18] can improve accessibility. Alternatively, solid walls can be removed completely while restricting different liquids to specific regions; this can be achieved by creating an additional aqueous phase using local concentrations of water-soluble polymers[15, 16], or by patterning surfaces with hydrophilic and hydrophobic patches[11–13, 19]. However, such approaches ultimately rely on constraining fluids with solid or polymeric walls, or pre-patterning surfaces.

Here we present a fundamentally different approach — Freestyle Fluidics (FF) — for handling small volumes that does not use solid walls or pre-patterned surfaces. FF circuits are created in seconds in much the same way as writing freehand with a pen. Just as any imaginable pattern can be drawn on a piece of paper, any fluidic circuit can be created by dragging a pen emitting liquid across an un-patterned substrate. Edges of circuits are pinned by interfacial tension, and liquids in them are confined by fluid walls; during flow, these walls adjust their shape above an unchanging footprint (i.e., the area in contact with the substrate). These circuits can be made with materials of proven biocompatibility — the culture media and polystyrene/glass dishes that biologists commonly use[20]. We demonstrate the platform by carrying out a number of microfluidic operations, including delivering drugs to human cells and triggering chemotaxis in bacterial biofilms. The versatility of this technology suggests it may open up a range of applications wherever small liquid volumes are manipulated.

## Results

**Method for creating circuits with fluid walls**. The method uses fluid walls to pin liquids to flat un-patterned substrates. To introduce the concept, a simple case is considered: a small drop of tissue-culture media is printed on a polystyrene tissue-culture dish using a syringe pump connected to a pen — a hollow stainless-steel dispensing needle or plastic tube — which is held by a 3-axis traverse just above the virgin substrate (Fig. 1a). Interfacial tension holds the drop in place, and its geometry becomes the cap of a sphere when gravity forces are negligible (Fig. 1b). This drop has fluid walls made of air and media. The physics governing wall shape when water is added or removed from a drop in air have received considerable attention[21, 22] (Supplementary Fig. 1 and Supplementary Note 1). Thus, for a given volume, the footprint depends on the equilibrium contact angle, $\theta_E$. In the case of the tissue-culture medium — RPMI — in

air, this angle ($\theta_{Ew}$) is ~50°. After printing the drop, slightly more medium can be added without the footprint increasing in area, with the exact amount being determined by the advancing contact angle. But once this angle is reached, footprint area increases (Fig. 1c). Fluid can also be removed without change in footprint (Fig. 1d) until the receding contact angle is reached. Here, the receding angle is so small that at least 95% of a 5-μl drop can be removed (and the contact angle falls to ~3°, calculated assuming drops are shaped like caps of spheres). This process allows fabrication of fluidic chambers that can accept more or less liquid without altering footprints.

To eliminate evaporation, drops are overlaid with an immiscible fluid. This can be less dense than media like a hydrocarbon; perhaps counter-intuitively, it can be denser like the fluorocarbon, FC40 — a transparent, fully fluorinated liquid (density 1.855 g/ml) that is widely used in droplet-based microfluidics. Instead of the aqueous phase rising due to buoyancy, interfacial forces dominate and the media remains stuck to the substrate. As the solubility of water in FC40 is <7 ppm by weight, a drop overlaid with FC40 open to air is stable for days at room temperature. FC40 also effectively isolates different drops from others in a dish, and from the surroundings, thereby preventing contamination. For example, an array of drops with and without bacteria are printed on a Petri dish; after overlaying FC40, bacteria grow in inoculated drops, as the rest remain sterile for 10 days on a non-sterile laboratory bench (Supplementary Fig. 2). Overlaying FC40 provides another useful property. At least 60% more fluid can be added to a drop before the footprint increases in area, as FC40 increases $\theta_E$ from ~50° to ~70° (Fig. 1e). The elimination of evaporation coupled to such

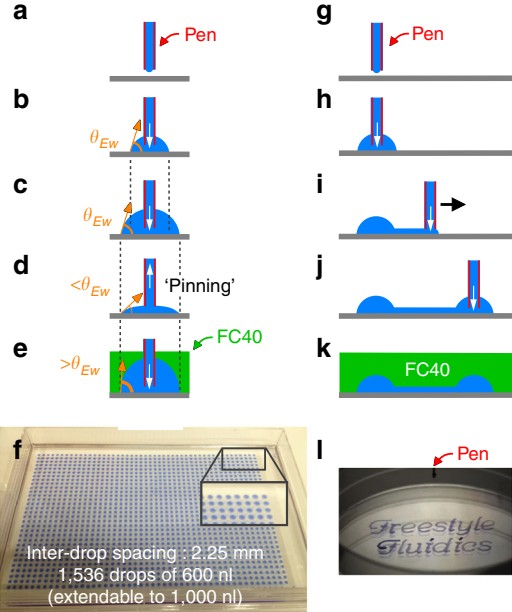

**Fig. 1** Making circuits without solid walls. **a–f** Some principles. $\theta_{Ew}$: equilibrium contact angle (media in air). **a, b** Ejected media is held in place by surface tension. **c** Adding media increases the footprint, but **d** removing large amounts does not. **e** Overlaying FC40 allows more media than in (**c**) to be added without altering the footprint because the equilibrium contact angle of media under FC40 is greater. **f** Array of 1536 drops of media plus blue dye under FC40 in a flat microtiter plate lacking wells (*inset* shows magnification). The pen ejected fluid continuously as it deposited drops (locations as in a conventional 1536-well plate). **g–k** Steps in printing a simple circuit. See text. **l** Example circuit printed in ~40 s in air using media mixed with blue dye in a 6-cm tissue-culture dish. It is not yet overlaid with FC40

isolation means that drops and arrays can be used as alternatives to conventional wells and plates with the advantage that working volumes are lower; aqueous liquids are simply pipetted into (and removed from) drops through FC40 instead of air (Fig. 1f).

FF circuits are constructed much like drawing on a piece of paper — the pen is moved as it continuously ejects media to deposit the required pattern on the substrate. Put in another way, if a circuit can be drawn on paper, it can be created in seconds using FF. Figure 1g–k illustrates fabrication of a simple circuit. The pen tip is brought close to (but not touching) the substrate, it deposits a drop/chamber, traverses in a straight line above the substrate to leave a trail of fluid (which becomes a conduit), and finally stops to deposit a second drop/chamber. Complex patterns can be printed in seconds (Fig. 1l; Supplementary Movie 1). As before, overlaying FC40 prevents evaporation. Alternatively, circuits can be printed under FC40; then, conduits have narrower footprints due to reduced spreading of the aqueous phase (Supplementary Fig. 3), but these are not discussed further here. FF circuits overlaid with FC40 are also stable for days, and pinning lines are strong enough to survive violent agitation (Supplementary Movie 2). These results illustrate how circuits can be prototyped quickly using little more than a pump and 3-axis traverse.

**Passive pumping**. Many existing microfluidic devices use external pumps to drive flow; this adds complexity and cost, and limits the number of devices that can be operated simultaneously. In contrast, flow through FF circuits can be driven passively without additional equipment. The principle used has been demonstrated previously: flow rate and direction may be controlled by varying Laplace and hydrostatic pressures[23–26]. Laplace pressure is given by $2\gamma/R$, where $\gamma$ is interfacial tension, $R$ is radius of curvature; hydrostatic pressure is $\rho g h$, where $\rho$ is density, $g$ is gravity, and $h$ is height. Thus, if two differently-sized drops of the same fluid are connected by a conduit and Laplace pressure dominates, the one with the smaller radius of curvature harbors a larger pressure; this drives flow from the small drop to the larger one (Fig. 2a). In Fig. 2b and Supplementary Movie 3, different volumes of red dye are pipetted into left-hand source drops – the smaller the source drop, the higher the flow to the right-hand sink (see Supplementary Note 1, and Tables 1 and 2 for details). During flow, footprints do not change, although volumes above footprints do (Supplementary Movie 4). Here, diffusional transfer to sinks can be neglected as distances are so long and time scales so short.

Varying conduit width provides another way of controlling flow. For example, when 18-µl drops are connected to 20-µl drops by conduits with different widths, flow is fastest through the widest because it has the lowest hydrodynamic resistance; flow rates vary over two orders of magnitude (~0.3-30 µl/h) when conduit width changes ~3-fold (Fig. 2d). With narrow conduits, flow remains steady for hours (Fig. 2d; Supplementary Fig. 4a, b illustrate a side view of a sink drop, and reproducibility of flows obtained with two identical circuits).

Interplay between Laplace and hydrostatic pressures dictates the shape of curves in Fig. 2d. If Laplace pressure were the sole driver of flow, the rate of volume reduction would progressively decrease with time. However, this reduction is counteracted by the changing hydrostatic pressure of the denser overlay. Supplementary Fig. 4c and associated text illustrate this interplay, and Table 1 provides example geometric data for isolated drops of different sizes (e.g., with a 3-mm overlay, a 5-µl drop at an equilibrium contact angle of 70° has a drop height, base radius, and base area of 1.1, 1.58 mm and 7.8 mm², respectively). Thus far, flow has been from the drop with the smallest footprint, but it can be in the opposite direction if the drop with the smallest

footprint radius has a larger radius of curvature[26]. This can be achieved using a flat drop (i.e., one with a contact angle much less than $\theta_E$; Fig. 2c). Overall, flow through a circuit is simple to

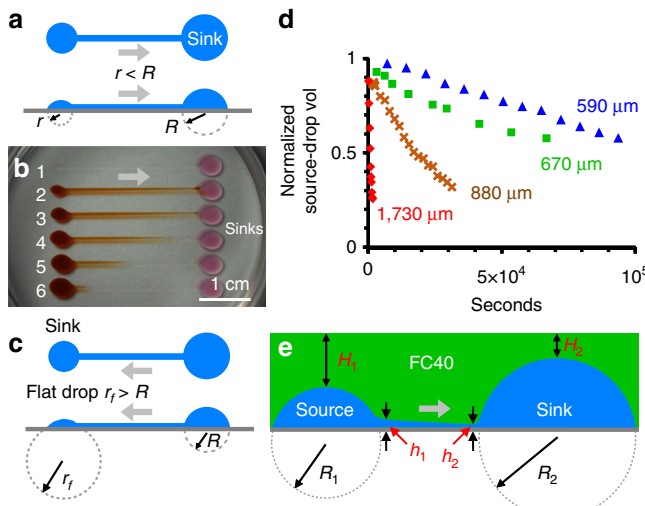

**Fig. 2** Using Laplace pressure to drive flow through FF circuits. *Grey arrows*: direction of flow. **a** Principles. In this simple circuit (plan, side views), the left-hand drop has the smaller radius of curvature ($r < R$), and a pressure difference between the drops is the main driver of flow. **b** A frame from Supplementary Movie 3. Media (20 µl) was added through the 5 ml overlay of FC40 to each sink drop; next, 10, 8, 6, 4, and 2 µl red dye were pipetted successively into drops 6–2, and the dish photographed after ~30 s. Advection transports dye away from drops with the smaller radii of curvature (diffusion down the conduit is negligible). Although dye was added to drop 2 last, it reaches a sink first. **c** This circuit (plan, side views) has the same footprint as that in (**a**), but flow is reversed because the small flat drop has the larger radius of curvature ($r_f > R$) and lowest pressure. **d** Flow rate depends on conduit width. Four circuits were made like the one in (**a**); each had an 18-µl left-hand (source) drop connected to a 20-µl sink through a 11-mm conduit (widths indicated). Circuits were overlaid with 3-mm FC40. Time-lapse imaging (side views) show volumes of source drops decrease over time; these volumes were determined and normalized relative to initial ones. **e** Interplay between Laplace and hydrostatic pressures. The left-hand drop has a smaller radius of curvature (and so higher Laplace pressure) than the right-hand one ($R_1 < R_2$), and is overlaid with a greater depth of FC40 ($H_1 > H_2$) and so experiences a higher hydrostatic pressure. Both pressures combine to drive flow to the right

### Table 1 Hydrostatic and Laplace pressures associated with isolated drops

| Parameter | Drop volume (µl) | | | |
|---|---|---|---|---|
| | **2.5** | **5** | **10** | **20** |
| Drop height (mm) | 0.88 | 1.10 | 1.39 | 1.75 |
| Base radius (mm) | 1.25 | 1.58 | 1.98 | 2.50 |
| Base area (mm²) | 4.91 | 7.79 | 12.37 | 19.64 |
| Hydrostatic pressure, FC40 (Pa) | 38.56 | 34.43 | 29.23 | 22.67 |
| Hydrostatic pressure, water (Pa) | 8.59 | 10.82 | 13.63 | 17.17 |
| Laplace pressure (Pa) | 60.13 | 47.73 | 37.88 | 30.07 |
| Pressure at base of drop (Pa) | 107.28 | 92.98 | 80.74 | 69.91 |

Drops have volumes specified and sit on a flat substrate (values calculated assuming a contact angle of 70°, an interfacial tension of 40 mN/m, and an overlay of FC40 with a depth of 3 mm)

**Table 2 Geometric parameters of drops and conduits**

| Conduit width $w$ (µm) | Parameter | Drop volume (µl) | | | |
|---|---|---|---|---|---|
| | | 2.5 | 5 | 10 | 20 |
| 300 | Radius of curvature, $R$ (µm) | 757 | 1038 | 1521 | 2586 |
| | Height of center of conduit, $h$ (µm) | 15.0 | 10.9 | 7.4 | 4.4 |
| | Contact angle, CA (degree) | 11.4 | 8.3 | 5.7 | 3.3 |
| | Length of interface, $L$ (µm) | 302.0 | 301.1 | 300.5 | 300.2 |
| | Cross-sectional area, CSA (µm²) | 3008 | 2181 | 1483 | 871 |
| 600 | Radius of curvature, $R$ (µm) | 757 | 1038 | 1521 | 2586 |
| | Center height conduit, $h$ (µm) | 62.0 | 44.3 | 29.9 | 17.5 |
| | Contact angle, CA (degree) | 23.3 | 16.8 | 11.4 | 6.7 |
| | Length of interface, $L$ (µm) | 616.9 | 608.7 | 604.0 | 601.4 |
| | Cross-sectional area, $A$ (µm²) | 25,001 | 17,795 | 11,974 | 6989 |
| 900 | Radius of curvature, $R$ (µm) | 757 | 1038 | 1521 | 2586 |
| | Center height conduit, $h$ (µm) | 148.3 | 102.6 | 68.1 | 39.5 |
| | Contact angle, CA (degree) | 36.5 | 25.7 | 17.2 | 10.0 |
| | Length of interface, $L$ (µm) | 963.8 | 930.9 | 913.7 | 904.6 |
| | Cross-sectional area, CSA (µm²) | 90,856 | 62,200 | 41,038 | 23,711 |

Conduits with varying conduit width are connected to drops of varying volume (values calculated assuming a contact angle of 70°, an interfacial tension of 40 mN/m, and an overlay of FC40 with a depth of 3 mm)

generate and can be controlled flexibly without using external pumps.

**Fluid walls adapt during flow**. In conventional devices, channels have fixed cross-sections; in contrast, heights of FF conduits adapt in response to changing pressures above unaltered footprints. If the cross-sectional area along the conduit in Fig. 2e is approximated by a segment of a circle, then its radius of curvature specifies the Laplace pressure across the interface (Supplementary Fig. 5). To a first approximation, curvature at each end of the conduit ($R_{conduit}$) may be obtained by assuming pressures at bases of source and sink drops are equal to pressures in the conduit near inlet and outlet respectively. The pressure drop across the conduit interface, assuming conduit height is small relative to drop height, is (Eq. 1; see Supplementary Note 1 for details):

$$\Delta P_{interface} = \frac{\gamma}{R_{conduit}} = \frac{2\gamma}{R_{drop}} - \Delta\rho_{(FC/water)}gh_{drop} \qquad (1)$$

Because conduit height decreases with local pressure, the cross-sectional area of the conduit is predicted to decrease in the direction of flow. Table 2 provides some examples; a conduit (width 600 µm) connecting 5- and 10-µl drops has center-line heights at inlet and exit (where heights are $h_1$ and $h_2$) respectively of ~44 and 30 µm, contact angles of ~17 and 11°, and cross-sectional areas of ~18,000 and 12,000 µm². As liquids are incompressible, volumetric flow rate must be the same at each cross-section along the conduit. Therefore, mean flow velocity increases in the direction of flow as cross-sectional area decreases, and – in this example – continuity dictates that it increases ~1.3-fold from conduit inlet to exit as fluid walls change their shape accordingly. Here, effects of gravity on conduit and drop interfaces are ignored.

**Example circuits using passive flows**. Exemplary circuits carrying out basic functions are now illustrated (Fig. 3); PDMS chips performing analogous ones have been described[2, 3, 27]. Colored dyes are being pumped passively through these circuits, and shapes of walls are determined by well-established principles (Supplementary Fig. 5). For example, conduits have widths mainly determined by pen width (and to a lesser extent by

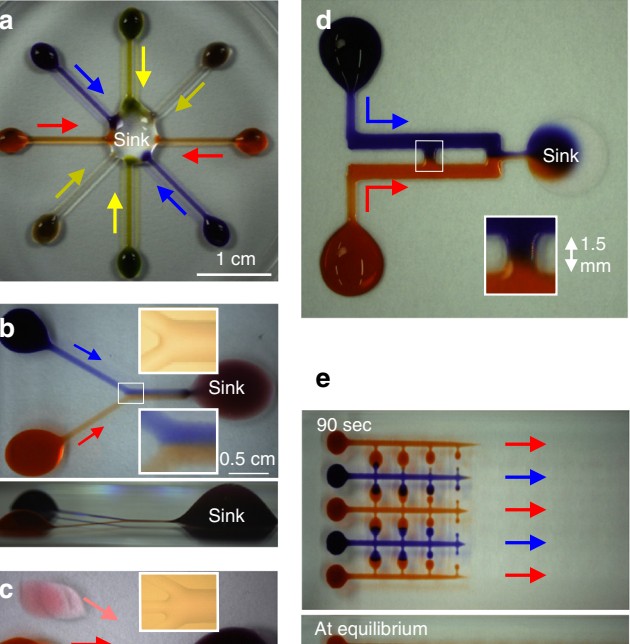

**Fig. 3** Some FF circuits carrying out different functions. Colored dyes were pipetted manually into input drops, and they are flowing (*colored arrows*) to sink drops autonomously. *Insets* illustrate how accurately pinning lines are built. **a** Mixing 8 fluids. **b** Generating a stable concentration gradient across two laminar streams after the junction (*side view below*). **c** Flow focusing the central laminar stream after the junction (*side view below*). **d** Generating a flow-free diffusion gradient across the central conduit (*inset*). **e** Feeding circuit after 90 s (*upper*) and at equilibrium (*lower*); dyes were pipetted into large drops on the left, which then feed the small chambers

ejection rate, contact angle, and pen-to-substrate distance), and heights along the center-line are typically down to a few microns (Table 2). Drop-like features are >4 mm wide so microliter volumes can be manually pipetted into them easily, but they can initially contain as little as ~100 nl. These circuits mix fluids (Fig. 3a, Supplementary Movie 5; Supplementary Movie 6 illustrates gravity-driven splitting of a stream), generate chemotactic gradients (Fig. 3b), flow-focus laminar streams (Fig. 3c, Supplementary Movie 7 and Supplementary Fig. 6 illustrates re-use of this circuit after switching two inputs), create a diffusion-based concentration gradient across a flow-free conduit (Fig. 3d), and feed many chambers from large inlet ports on the left (Fig. 3e). This collection of experiments demonstrates the versatility of the platform.

**Human cells grow normally in FF drops and circuits.** These circuits can be constructed using biocompatible liquids (i.e., tissue-culture media), substrates (i.e., polystyrene Petri dishes/ glass slides), and overlay (FC40 is bio-inert, permeable to the vital gases – $O_2$ and $CO_2$, and small molecules secreted by cells and/or added drugs are less likely to partition into this fluorinated oil compared to a conventional hydrocarbon oil, PDMS channel, or aqueous biphasic system). To test bio-compatibility, human embryonic kidney (HEK) cells were plated in a drop of DMEM plus serum on a standard tissue-culture dish, and the dish overlaid with FC40 and mounted on a microscope in an atmosphere of 5% $CO_2$; imaging shows cells grow like their counterparts cultured conventionally (Fig. 4a; Supplementary Movie 8).

HEKs also respond normally to a drug. This is illustrated using a circuit that autonomously creates serial dilutions. This circuit is made and operated as follows. In Fig. 4b, chambers 1, 2, and a–f are printed as concentric sets of 3 circular conduits spaced less than one pen-diameter apart; these fuse to form single flat chambers with identical footprints, shapes, and pressures. Pipetting dyes into 1 and 2 now increases pressures locally, which drives flow into a–f; then, a fills only with red dye, f only with blue dye, and b–e with dilutions of the two (b ends up with the most red dye and the least blue, and e with the most blue dye and the least red). These HEKs had been genetically-modified to encode a GFP-reporter gene controlled by a promoter switched on by tumor necrosis factor alpha (TNFα) – when exposed to the cytokine, they fluoresce green. Therefore, when cells are seeded in lettered chambers, grown in a standard $CO_2$ incubator for 18 h, and then TNFα pipetted into 1 and medium into 2, the circuit now serially dilutes TNFα to give the highest concentration in a, and dilutions in b–e (f receives no TNFα). Note that advection creates the concentration gradient, and the concentration of TNFα changes by less than 0.5% over 24 h due to subsequent diffusion (Methods). After regrowth to allow TNFα to switch on GFP expression, imaging reveals that cells respond in the expected range of cytokine concentrations; GFP fluorescence is highest in a, with intensity tailing off to background in f (Fig. 4c). These results support the biocompatibility of our circuits, and illustrate a circuit that serially dilutes drugs autonomously.

**Flow driven by an external pump.** Many microfluidic applications require stable flows persisting for days[27]; this is difficult to achieve by passive pumping because flow rates inevitably change over time (Fig. 2b). One application – chemotaxis – requires stable flows, and many microfluidic devices have been developed to study it[27]. Therefore, we sought to establish stable flows by incorporating external pumps into a Y-shaped chemotaxis circuit. After printing (Supplementary Movie 9), tips of two stainless-steel dispensing needles – filled with blue or red dye and connected to syringes mounted on one pump – were lowered through

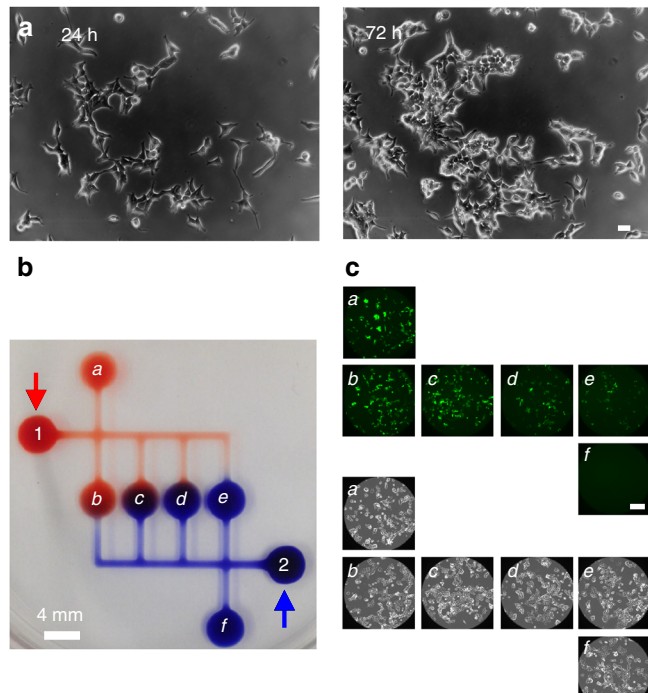

**Fig. 4** HEKs in drops/circuits in 6-cm plates grow as expected. **a** Phase-contrast images (frames from Supplementary Movie 8) showing cells in an FF drop increase in number. *Bar*: 40 μm. **b** Circuit operation demonstrated using dyes. *Blue* and *red* dyes were pipetted into drops 1 and 2 (*arrows*); they flow autonomously into chambers a–f to create serial dilutions in minutes (a contains the highest concentration of red dye and no blue, while f contains the highest concentration of blue dye and no red). **c** Cells in the FF circuit respond to TNFα. A total of 1 μl HEKs (~600 cells) were plated in each chamber a–f, grown (24 h), and TNFα (9 μl; 10 ng/ml) pipetted into drop 1 and medium (9 μl) into 2. Automatic dilution/mixing gives the highest concentration of TNFα in a, and serial dilutions in b–e; f receives no TNFα. TNFα concentrations in chambers a–f were 5.1, 4.7, 3.4, 1.8, 0.8, and 0 ng/ ml. Cells were now incubated for 24 h to allow TNFα to induce GFP expression. Fluorescence (*upper*) and bright-field images (*lower*) of the centers of chambers a–f are shown. For quantitative analysis of TNFα concentrations and fluorescence intensities, see Supplementary Fig. 7. *Bar*: 200 μm

FC40 until they pierce an arm of the Y; the aqueous interface then spontaneously seals around the hydrophilic needles (Fig. 5a, Supplementary Fig. 8). On starting the pump, red and blue dyes are injected into the circuit and flow around right-angle bends at rates up to ~1 ml/h without changing the footprint. If air bubbles are introduced via inlets, their buoyancy forces them to pinch off and be lost to the atmosphere (Supplementary Movie 10). These results confirm that external pumps can be connected simply through self-sealing gaskets to FF circuits, fluid walls robustly adapt to changing flows, footprints remain unchanged over 9 h (longer times can be accommodated by removing fluid from the sink by pipette, or using a flat sink of larger diameter), and circuits can be operated like their counterparts embedded in PDMS without problems associated with leaky seals and air bubbles.

**Bacterial chemotaxis in developing biofilms.** We next ascertained whether previous results obtained using a conventional PDMS-based device could be replicated using this chemotaxis circuit. Biofilms play important roles in many human infections and environmental processes[28]. Recently, it was shown that individual bacteria in biofilms growing on glass in PDMS can

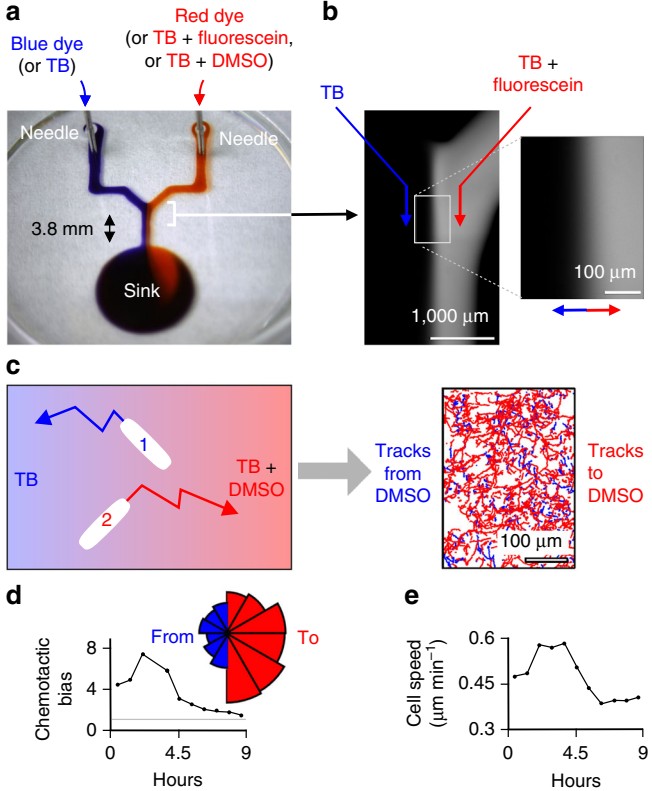

**Fig. 5** Integrating external pumps into an FF circuit to study bacterial chemotaxis. **a** Overview (40-mm glass-bottomed dish). Blue and red dyes (or alternatives indicated) each flow (100 nl/s) from syringes driven by one external pump through hollow needles to the sink. **b** Using fluorescein to characterize the diffusion gradient across the conduit. The circuit was placed on a confocal microscope, and TB and TB + fluorescein were pumped (12 µl/h) through arms of the Y. An image of the region downstream of the junction reveals fluorescent and dark laminar streams. *Inset*: diffusion of fluorescein to the left generates a concentration gradient (*arrows* point to highest concentrations). **c** Chemotaxis of *P. aeruginosa* towards DMSO. The circuit was placed on an inverted microscope, bacteria pipetted into the central conduit, TB and TB + DMSO injected into left and right arms (each at 12 µl/h), bright-field images collected over 9 h, and trajectories of individual bacteria in a region near the junction determined. The cartoon (*left*) shows individual trajectories (cells 1 and 2 move down and up the gradient, respectively), and the map (*right*) shows more trajectories (*red*; collected between 0 and 6 h) are to the right towards high DMSO concentrations compared to those to the left (*blue*). **d** Chemotactic bias (number bacteria travelling up DMSO gradient divided by number moving down). Bias > 1 indicates more cells move up gradient (*grey line*: lack of chemotaxis). *Inset*: probability-density functions of angle from each trajectory's origin to final position (0–6 h), with *red/blue* bins denoting movement towards/away DMSO. A slight downstream bias occurs because flow pushes bacteria. **e** Average speed of individual cells increases initially before decreasing due to cell crowding caused by population growth

sense chemical gradients and move towards nutrients[29]. While bacteria in suspension swim at tens of body lengths per second by rotating helical flagella, on surfaces they use tiny grappling hooks — pili – to pull themselves ~1,000-fold more slowly[30–32]. As timescales are long and gradients steep, these experiments are challenging to perform in PDMS-based devices. Air bubble formation is particularly problematic; as bubbles travel through devices they detach cells from surfaces, unbalance flows, and alter gradients[7, 8].

Our chemotaxis circuit was initially created using tryptone broth (TB) on a glass-bottomed Petri dish; however, pinning lines

were unstable. Therefore, the circuit was printed using DMEM + 10% FBS, and then washed through with excess TB; pinning lines remained unchanged during washing, and subsequently during the experiment. This shows a circuit can be created with one fluid giving stable pinning lines, and operated using another. [An alternative involves treating glass to give stable pinning in TB, but this was less preferred because previous results had been obtained using unmodified surfaces.]

To characterize the chemotactic gradient, the Y-shaped circuit was mounted on a confocal microscope, and an external pump used to create two laminar streams of TB in the central arm of the Y – one labeled with fluorescein (used as a surrogate for a chemoattractant). Then, fluorescence imaging revealed a bright stream flowing side-by-side with a dark stream (Fig. 5b), and a diffusion gradient of dye across the conduit (Fig. 5b, *inset*). With slow flow, there is time for diffusion to create a gradient ~100 µm wide; with fast flow, there is less time and the gradient is steeper (Supplementary Fig. 8d).

For the chemotaxis experiment, pathogenic bacteria – *Pseudomonas aeruginosa* – were manually pipetted into the central arm of the Y; cells attach to glass. The two input needles now inject TB and TB plus the chemoattractant, dimethyl sulfoxide (DMSO); this washes away unattached cells and diffusion between laminar streams then creates the chemotactic gradient above the nascent biofilm. Cells on the substrate were motile (Supplementary Movie 11). Automated tracking algorithms[29] were now used to extract trajectories of > 10,000 cells; many more bacteria move towards DMSO than away from it (Fig. 5c; compare numbers of red and blue tracks). Both chemotactic bias and cell speed peak after a few hours and then decline as bacteria divide, and the resulting crowding attenuates movement (Fig. 5d, e). These results were similar to those obtained with the PDMS-based device[29] (Supplementary Fig. 8e). This indicates that FF circuits can provide analogous data to those obtained with conventional systems more rapidly and cost-effectively, with fewer technical drawbacks.

## Discussion

We have presented a microfluidic platform – Freestyle Fluidics (FF) – in which liquids drawn on flat un-patterned substrates are confined by fluid walls and ceilings (Fig. 1); these walls/ceilings change shape above the footprint when fluids flow through the system. This platform has many advantages. First, almost any imaginable 2D design can be printed cheaply in seconds using an aqueous solution and plastic/glass substrates. Fabrication does not require a dedicated microfluidics laboratory or specialized equipment beyond a syringe pump and an automated positioning system to drive the pen, and a laminar-flow hood if sterile circuits are required. Second, circuits are fully accessible from above. Consequently, micro-liter volumes can be pipetted manually into them at any point to provide a simple interface between micro- and nano-liter scales; smaller volumes can be pipetted into smaller features using automatic systems. Third, fluid walls are built accurately and reproducibly by interfacial forces (Fig. 3, insets). Importantly, these walls are strong, pliant, and resilient (Supplementary Movie 2); they self-heal and reform when breached. Fourth, shapes of fluid walls can be varied locally to create differences in Laplace and hydrostatic pressure that drive fluids passively through circuits (Fig. 2). As such pumping is simple and scalable, it can be applied to high-throughput analyses. Flow can even be reversed or stopped by adding/removing fluid from selected drops to alter local pressures. Fifth, external pumps are easily integrated into circuits if stable flows of large volumes are required for long times; tubes connected to a pump are lowered through FC40 until they pierce fluid walls, and then walls

spontaneously seal around inserted tubes (Fig. 5). Sixth, the method is biocompatible and especially useful for live-cell assays (Figs. 4 and 5 demonstrate human cells and bacterial biofilms growing and responding to stimuli as expected); it requires just a bio-inert fluorocarbon and the culture media and polystyrene/glass dishes used by biologists. Seventh, conventional circuits are often rendered non-functional by air bubbles, but if accidentally introduced into FF circuits, buoyancy forces them to rise to the surface without altering footprints (Supplementary Movie 10).

As with any method, FF has limitations. First, fluids and surfaces must be matched to ensure pinning lines are stable. Fortunately, suitable combinations can be screened rapidly by placing a drop of fluid on a substrate, and then removing most of the volume. If the pinning line does not retract, the combination may be used; if it retracts, a circuit can be created using a liquid known to allow fabrication and then washed through with the one desired for operation (as in Fig. 5). Second, many existing PDMS circuits cannot be replicated exactly, so new designs to achieve existing functions must be developed. For example, the fluid walls in FF circuits are curved and not straight, FF conduits have minimum lengths and widths larger than many PDMS channels (we typically use printing tips with diameters of hundreds of microns), source drops have maximal diameters of ~5 mm (giving an upper volume of ~20 μl for a contact angle of 70°), and pressures tolerated are lower because of the limitations imposed by interfacial forces (Supplementary Note 1). Third, liquids used in cell biology are often transparent, so FF circuits are difficult to see; therefore, we often print the circuit plan on paper, place it under the dish, and then use the plan as a guide when manually pipetting into circuits. Fourth, circuits should usually be horizontal during operation (achieved using a bull's-eye spirit level).

In summary, we have developed a versatile microfluidic platform for constructing and operating microfluidic devices that uses liquid interfaces. Circuits can be prototyped quickly; they can be made in much the same time that it takes to draw them by hand. The simplicity and flexibility of our method is designed to bring microfluidics to a wide range of laboratories and applications.

## Methods

**General reagents and equipment**. FC40 was purchased from Acota. It is bio-inert, and not found in regulatory lists of dangerous organic chemicals (http://www.acota.co.uk/assets/data-centre/msds/3m/3mfc40msds.pdf). If circuits are to be kept for days, extra FC40 should be added when needed, and we give the following data as a rough guide to the replenishment rate. The vapour pressure of FC40 is 432 Pa at 25 °C, so FC40 evaporates relatively slowly compared to water (vapour pressure of 3170 Pa). We find experimentally that the rate of evaporation of FC40 at 25 °C from a 6-cm Petri dish is 90 μl per day with the lid on, and 1.55 ml per day with the lid off. Evaporated FC40 also had no untoward effects on any of many different cell types grown conventionally in the same incubator at the same time over a period of 2 years. All other fluids and materials were from Sigma Aldrich unless otherwise stated. Where indicated, aqueous drops contained water-soluble dyes (e.g., 4 mg/ml Allura Red, 2 mg/ml toluidine blue). Circuits were generally printed on '6-cm' polystyrene tissue-culture dishes (Falcon; 60 × 15 mm style), which have an internal diameter of 5 cm, rectangular flat polystyrene micro-titer plates (127.7 × 85.5 mm; Nunclon from Thermo Fisher Scientific), or glass microscope slides and coverslips. Blunt stainless-steel dispensing needles used for pens generally had widths of 0.4–0.6 mm outer diameter.

**Cells**. *Escherichia coli* (chemically-competent TOP10 bacteria; ThermoFisher Scientific) used in Supplementary Fig. 2 were inoculated directly from frozen stocks into SOC medium (Invitrogen). A total of 50 × 1 μl drops containing bacteria were deposited on a 60 mm-dish, interspersed by 50 × 1 μl SOC-only drops. Drops were subsequently overlaid with FC40 and incubated as indicated.
HEK-293 reporter cells (NF-κB/293/GFP-Luc™ Transcriptional Reporter Cell Line; System Biosciences, catalogue number TR860A-I) were grown as recommended by the manufacturer in DMEM plus 10% FBS, and in FF drops and circuits overlaid with FC40 in exactly the same way (i.e., in the same medium, in a 5% CO₂ incubator at high humidity). They encode a GFP gene under the control of the minimal cytomegalovirus promoter downstream of four copies of the NF-κB

consensus transcriptional-response element. GFP expression was induced by treating cells with varying levels of TNFα (Peprotech) for times indicated. Results obtained in Fig. 4c were like those reported by the supplier of the cell line.
*Pseudomonas aeruginosa* PAO1 (Kolter collection, ZK2019[29]) was used for the chemotaxis assay (Fig. 5). Cells were grown from frozen stock overnight in 3 ml Luria–Bertani broth (LB Lennox, 20 g/l) at 37 °C with shaking at 250 rpm. Cells were subcultured (1:30 dilution) in TB (10 g/l Bacto™ tryptone, Becton Dickinson and Company) to obtain cells in exponential phase. Cells were then diluted in TB to an optical density at 600 nm of 0.25 before being pipetted into circuits. Circuits were incubated at 20 °C for the duration of the experiment. Results obtained in Fig. 5c–e were like those obtained with a commercial PDMS-based microfluidic system and PDMS-based devices fabricated in house[33].

**Measurement of interfacial tension**. Interfacial tensions were measured using the pendant-drop technique and a commercial system (First Ten Angstroms 1000). Drops were ejected from 16–30 gauge stainless-steel blunt needles using a programmable syringe pump (Harvard PhD Ultra I/W) or micro-meter syringe (Gilmont) into a less-dense fluid in a 2 ml cuvette. The manufacturer's software was used to calculate the interfacial tension for each image. Before using new fluids, the system was calibrated; the interfacial tension of filtered water/air or FC40/air was measured and good agreement was found with established values of 72 and 16 mN/m.

**Printing and operation of FF circuits**. To print the simple circuit in Fig. 2a, a blunt stainless-steel dispensing needle (outer diameter 0.5 mm) was connected via PTFE tubing to an air-tight glass syringe which was prefilled with culture medium (i.e., RPMI + 10% FBS) and mounted on a programmable syringe pump (Harvard PhD Ultra I/W). [This medium was used for all circuits unless stated otherwise.] The needle can be held vertically by an automated positioning system – a 3D traverse system (Z-400, CNC Step, Germany) or an integrated Freestyle printer (iotaSciences Ltd, UK). Beds of the system and printer were levelled using miniature bull's-eye spirit levels. The needle tip was brought to within ~100 μm of the surface of a horizontal 60-mm dish by first lowering the tip until it touched the surface and then raising it 100 μm. Next, the pump was started so the tip ejected fluid (300 nl/s; in other cases, ejection rates varied from 100 – 2,000 nl/s depending on the size of the needle tip) as it remained in a fixed position until a drop of desired size is formed, moved laterally (traverse rate 30 mm/s) to leave a trail of medium behind the substrate (the conduit), and remains stationary to form the second drop. At the end, the needle is retracted from the substrate, the pump stopped, and the needle moved to the new desired location if another circuit is to be printed. Complex circuits can either be printed using continuous flow (as in Fig. 2b and Supplementary Movie 1), or by stopping and starting the pump, and retracting the tip from the surface, as individual features in a circuit (and even parts of a feature) are made. To prevent evaporation, FC40 is poured into the dish to a sufficient height to cover the circuit. Prudence dictates that FC40 is poured next to (and not directly on to) a circuit, but experience indicates that pinning lines are usually strong enough to remain unchanged even when FC40 is poured directly onto circuits (Supplementary Movie 12 illustrates addition of FC40 to a circuit, and Supplementary Fig. 3g shows part of a circuit that was created while FC40 flowed over it).

The array of drops in Fig. 1f (spaced as in a conventional 1,536-well plate) was made by continuously ejecting (1,000 nl/s) medium plus blue dye from a needle (external diameter 0.6 mm). The needle was lowered to eject the first drop (600 nl), raised, moved laterally (traverse rate 30 mm/s), and lowered to eject the second, and so on. As pinning lines of drops remain unaltered over a wide range of contact angles, liquids can be added to, or removed from, a drop in the same way as a conventional well. However, each FF drop has a smaller working volume (~0.6–1 μl), compared to the ~2–10 μl in a conventional well. FF arrays yield another advantage if the final readout involves imaging. Contents of interest near the wall of a conventional well can only be imaged using a microscope directly from above because walls bring significant edge effects; in contrast, many FF drops can be imaged simultaneously from one point of view because FC40 is optically transparent (Fig. 1f, inset). Moreover, even more drops can be packed into the same area by reducing drop-to-drop spacing or drop volume (e.g., geometrical considerations indicate that ~1,855 0.1-μl drops can be packed into the standard area even with a generous inter-drop dead space of 1-drop diameter). FF drops with footprints of any shape (e.g., square, hexagonal) can also be printed to maximize use of the surface.

For Fig. 3a, 5-μl drops feed a 30-μl sink, the circuit was created in a way analogous to those in Fig. 2, and was not overlaid with FC40.
The Y-shaped circuit in Fig. 3b was made by ejecting medium (600 nl/s; 0.61-mm needle) on to the plastic. The tip was held stationary to create a left-hand drop (1 μl), moved to form a dog-leg channel (footprint 540–570 μm wide), and held stationary to create part of the sink drop (1 μl). An identical and reflected second circuit was now created below the first offset by 500 μm, so the two parts of the sink drop overlap, and ends of dog-legs run side-by-side (and merge to give a combined width of 1030 μm). The circuit was overlaid with FC40 (depth 4 mm). To start flow, 10 μl blue dye, 10 μl red dye, and 20 μl medium were hand-pipetted into the two left-hand drops and sink, respectively. Consequently, the sink initially contains roughly twice the volume of left-hand drops. After the junction, both dyes diffuse

across the laminar interface; this creates concentration gradients perpendicular to, and in the direction of, flow.

For Fig. 3c and Supplementary Movie 7, the circuit was made as in Fig. 3b with an additional section containing an input drop (the middle one on the left) and a partial sink drop (both of 1 μl) connected by a straight channel. Final widths of channels before and after the junction are 590–610 and 1,550 μm. To start flow, 10 μl medium, 10 μl red dye, 10 μl blue dye, and 20 μl medium were pipetted into left-hand drops and sink, respectively. After the junction, the central laminar stream (red) is flow-focused due its higher velocity and greater height of the conduit along the center-line. This circuit was reused in Supplementary Fig. 6.

Figure 3d illustrates another circuit for chemotaxis, but a (diffusion-based) concentration gradient is stably maintained across the central conduit (width 1 mm, length 1.5 mm; see inset) in the absence of flow through that conduit; however, flow sustains a constant dye concentration at the top and bottom of this conduit. Inspection shows that some blue dye passes down through this conduit as red dye passes up – demonstrating that transfers result from diffusion (not advection). The flat sink drop (diameter 6.5 mm) was made by printing a set of concentric circles; it has a smaller footprint than left-hand source drops (diameters 3.3 mm). The circuit was drawn (0.5 mm needle; traverse rate 20 mm/s; flow rate 300 nl/s) in air on a rectangular glass coverslip for improved imaging, and not overlaid with FC40. Conduits are 0.7 or 1 mm wide. Initially, 6.5 μl blue and red dyes were pipetted into left-hand drops; then, fluids flow along the shortest route to the flat sink drop on the right (volume ~1 μl); this flow ensures a (relatively) uniform concentration of blue (or red) dye is found at the top (or bottom) of the central conduit.

For Fig. 3e, the circuit was made in air in a flat micro-titer plate, a 0.5-mm needle, and a flow rate of 350 nl/sec. This circuit allows feeding of culture medium from inlet ports (the large left-hand chambers) to an array of small flat chambers spaced as in a 384-well plate. This circuit could be used to deliver fresh medium to each small chamber without change in footprint, and without any liquid flowing from one small chamber to another; when flow ceases, pressure must be constant everywhere in the circuit, and therefore each chamber must have an equal volume if wetted areas are similar. The five input chambers (left) have circular footprints (~3.4 mm diameter) and are connected to 5 main conduits (footprint widths ~1.1 mm). There are also 50 small flat chambers (only a maximum of 40 can be seen in fields shown) with circular footprints (3.4 mm diameter) connected to the main conduits through smaller feeder conduits (width ~0.6 mm). The small chambers are spaced 4.5 mm apart. Once printed, the dish was filled with 6 ml FC40, and flow initiated by pipetting 10 μl red or blue dye into input chambers. Then, the system equilibrates over ~40 min. As all small chambers have the same footprints, the requirement that the pressure is similar throughout the network at equilibrium ensures their final volumes are the same. Consequently, of the 10 μl deposited in each input chamber, only 3.3 μl remains at equilibrium (and each small chamber increases in volume by ~670 nl). One image in Fig. 3e was taken 90 s after adding red/blue dye to a large drop; equilibrium was reached after ~40 min when dyes fill all small drops. The other image in Fig. 3e was collected after 12 h.

In Fig. 4b, c, circular chambers have footprint diameters of 4.2 mm, the center-to-center distance between each chamber in the series b to e is 4.5 mm, conduits from source chambers 1 and 2 are 1.1 mm wide, and those feeding lettered chambers are 0.8 mm wide (see Supplementary Note 3 for the G-code used). The circuit was made using a 0.5 mm hollow stainless-steel needle as it traversed (20 mm/s) emitting a total of ~3 ml medium (flow rate 100 nl/s) on to a 60-mm dish. All chambers were identical, and each was made by printing concentric circles which fused together to give a flat chamber. 1 μl DMEM media + 10% FBS with cells (600 cells/μl) were pipetted manually into chambers a–f; then, the dish was placed in a $CO_2$ incubator. After 24 h, 9 μl media + TNFα (10 ng/ml) was added to source drop 1, and 9 μl media to source drop 2; this increases the Laplace pressure in drops 1 and 2, and fluid is passively pumped into chambers a–f. When flow by advection stops, pressures in all chambers are equal, so end volumes (i.e., ~3.6 μl, determined by manually aspirating all fluid from chamber by pipette) are also equal (assuming interfacial tension is equal at the interface). The result is a concentration gradient of the single drug. The initial transport of TNFα from source drop to cell chambers is by advection; once advection ceases, diffusion becomes the mass transport mechanism. Therefore, two questions arise: how far does TNFα diffuse, and by how much does diffusion change concentrations during the experiment? To provide theoretical answers, we use the diffusion coefficient (D) of a typical protein – the green fluorescent protein (molecular weight 27 kD) – in water (i.e., ~1 × 10⁻⁶ cm²/s)[33]. The diffusion distance, x, of this protein over 24 h (t) is 4.15 mm (calculated using the established relationship $x = \sqrt{2Dt}$). Therefore, the green fluorescent protein would diffuse ~4.15 mm from a chamber into a conduit in 1 day, and take ~11 days to diffuse from one chamber to another. Consequently, there will be no diffusional transfer of proteins between cell chambers during our experiment. However, the concentration in a chamber may change over time due to diffusion into a conduit, and this change can be calculated using theory provided in Supplementary Note 1. The volume of a conduit in Fig. 4b that connects cell chambers (length 4.15 mm) is 0.03 μl, compared to a chamber volume of 3.6 μl. If the average concentration over this diffusion length is half that in the chamber, then only 0.4% of a protein will be lost from the chamber to the conduit in 24 h.

For Fig. 5 and Supplementary Movies 9–11, circuits were printed in air on a Nunc Glass Base Dish (outer diameter ~40 mm; viewing-area diameter 27 mm; Thermo Scientific) using DMEM + 10% FBS, overlaid with FC40, and two

stainless-steel needles (0.6 mm diameter) inserted into the two inlet arms (Fig. 5a). Circuits were then washed through with >20-fold more TB than the DMEM that was used to construct the circuit. For Fig. 5c–e, 0.5 μl P. aeruginosa in exponential phase were pipetted into the central arm of the Y downstream of the junction. The two inlet tubes were loaded with either TB or 350 mM DMSO + Chicago Sky Blue 6B dye (0.03 mg/ml) in TB. This dye does not induce chemotaxis in P. aeruginosa[29]. The two inlet needles infused both liquids into the circuit (flow rate in Fig. 5 was 12 μl/h, but flows up to 300 μl/h have also been used with this circuit without altering pinning lines during the initial washing). Next, most fluid pumped into the sink was removed manually from the sink by pipetting. Gradients of DMSO (or dye) form downstream of the junction perpendicular to the direction of flow. Unattached cells are washed away to the sink, and remaining attached cells were imaged for 9 h in two adjacent, non-overlapping fields of view centered on 1186 and 1,676 μm downstream of where the two streams meet.

When printing circuits under FC40 (as in Supplementary Fig. 3), the pen tip is brought closer to the surface compared to printing in air and is typically ~50 μm away.

**Imaging**. Images of circuits lacking cells were collected using a zoom lens and digital SLR camera (Olympus D7100 DSLR) connected to an epi-fluorescent microscope (Olympus IX53; 1.25×, 4×, 10×, 25× objectives) with translation stage and overhead illuminator (Olympus IX3 with filters) for bright-field images.

Bright-field, phase-contrast, and fluorescence images of FF drops containing cells were collected using a camera (AxioCam MRm) attached to a microscope equipped for live-cell imaging (Zeiss Axioskop 40; Olympus LWD A20 PL 20× lens). For the live-cell movie of HEKs (Supplementary Movie 8), cells were plated in a 4-μl drop (150 cells/μl) and overlaid with FC40, and grown for 24 h before imaging began; then, images were taken every 10 min for 48 h using a Zeiss Axiovert 200 microscope with a Photometrics Coolsnap HQ Camera. TIFF-image files were merged using Metamorph software (Molecular Devices) and converted to AVI-format (7 frames/s) using ImageJ[34]. The final concentrations of TNFα in chambers a–f in Fig. 4c were determined using the fluorescence intensity of a surrogate – fluorescein isothiocyanate isomer I (0.5 mg/ml; Sigma Aldrich). For Supplementary Fig. 7c, and after growing cells for 24 h in TNFα, bright-field and fluorescence images of chambers a–f were captured using a CCD camera system (Ascent A16000, Apogee Imaging Systems) attached to the Axioskop-40 microscope (Zeiss; A20PL phase-contrast objective 0.40 160/1.2 from Olympus). Fluorescence intensity and cell area were quantified using ImageJ. Thus, for the cell area, bright-field images were analyzed using a customized plugin which dilates the outline of each cell (or cells) in contact; for fluorescence, the integrated density for the whole image was quantified. Intensities in the bar chart represent arbitrary values of the integrated density (fluorescence)/total cell area. For Fig. 5b and Supplementary Fig. 8d, and to quantify chemical gradients within the circuit and how they change as a function of imposed flow rate, experiments were performed without cells; fluorescein (0.5 mg/ml) was again injected through one inlet, and the region below the junction imaged (where diffusion between laminar streams creates a dye gradient) using scanning-laser confocal microscopy (Zeiss LSM 700 confocal system, Zeiss EC Plan Neofluar 10× objective). The data in overlapping fields of view were collected, which were then combined to generate a single stitched image that extends the length of the channel. Parts of stitched images are shown in Fig. 5b and Supplementary Fig. 8d. For the chemotaxis assay (Fig. 5c–e), images of surface-attached P. aeruginosa were captured at a frame rate of 1 frame/min using a Zeiss Axio Observer inverted microscope with a Zeiss Plan-Apochromat 20× objective, Zeiss AxioCam MRm camera, and a Zeiss Definite Focus system. The Chicago Blue dye mixed with DMSO was used to locate the position of the gradient. However, a weak concentration of dye was intentionally used so as not to adversely affect automated cell tracking. Subtle variations in background of the bright-field image caused the dye concentration to appear slightly uneven (the dye was post-processed to appear *red* in Supplementary Movie 11).

**Analysis of cell trajectories during chemotaxis**. To characterize the chemotactic response in Fig. 5c–e, >10,000 cell trajectories were measured using the TrackMate plug-in for Fiji[29, 35, 36], which were then post-processed using Matlab. Consistent with its name, twitching motility in P. aeruginosa is inherently unsteady, such that a cell's instantaneous direction of movement is not consistent with the mean direction that it moves over longer periods[32]. Therefore, trajectories were categorized as moving up or down the DMSO gradient using their net displacement over the entire trajectory. The number of cells moving towards larger concentrations of DMSO was divided by the number of cells moving towards smaller concentrations to calculate the chemotactic bias, which quantifies the strength of the response[37]. Both non-motile cells and those which do not move significantly from their initial position were eliminated from this analysis by excluding trajectories whose net to gross-displacement ratio (NGDR[38]) was <0.15. Here, NGDR is defined as the straight-line distance between a trajectory's beginning and end, divided by the gross distance a cell moves over the same time period. Finally, the angle between a trajectory's start and end position was calculated in order to generate the rose-plot (Fig. 5d, inset); these angles were pooled across all trajectories over the first 6 h of the experiment (in which chemotaxis was most pronounced), and binned into 12 equal intervals around the unit circle.

**Data availability**. The data that support the findings of this study are available from corresponding authors upon reasonable request.

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

## Acknowledgements

We thank the European Commission for a 7th Framework Marie Curie Career Integration Grant contract No. 333848 (E.J.W.), the Impact Acceleration Account of the Biotechnology and Biological Sciences Research Council (E.J.W., P.R.C.), the Medical Research Council for a Confidence in Concept award MC_PC_15029 (E.J.W., P.R.C,) and award MR/K010867/1 (P.R.C.), iotaSciences Ltd (A.F., A.N.T.), the Human Frontier Science Program for Fellowship LT001181/2011L (W.M.D.), the University of Sheffield for a starting grant from 'Imagine: Imagining Life Initiative' (W.M.D.), the EPSRC for a pump-priming grant to Sheffield University to study 'Anti-microbial Resistance' (W.M. D.), Magdalen College, Oxford, for a grant from the Calleva Research Centre for Evolution and Human Science (K.R.F.), and the European Research Council for grant 242670 (K.R.F.).

## Author contributions

E.J.W, A.F., W.M.D., K.R.F., and P.R.C.: Designed research, A.F., J.H.R.W., A.N.T., W.M. D., K.R.F., and E.J.W.: Performed experiments, and all authors wrote the paper.
