## [Peer Review File · Nature Communications]

Reviewers' comments:

Reviewer #1 (Remarks to the Author):

The authors present an innovative approach to “fabricating” fluidic channels by simply patterning an aqueous phase (i.e., without channel walls); an immiscible fluorinated oil is used to “contain” the aqueous phase and prevent evaporation. A key advantage of this method is the ability to pattern the channels on any surface, including standard commercially available cell culture surfaces. Another main advantage is the simplicity of the approach. Both of these advantages are well described in the introduction with citations to past literature motivating the present method. While the concept of pinning and using virtual walls in microfluidics is not novel (and the authors correctly cite past literature), the simple embodiment shown here and the numerous applications and tests are noteworthy and significant. These include showing applicability with passive and active pumping, multiple cell culture experiments, creating fluidic networks, robustness to evaporation and mechanical shaking, resistance of cultures to contamination on the open bench, resistance to leaking at tubing interfaces, and mitigating the need to troubleshoot air bubbles. I think this work will be interesting to a broad audience including researchers in microfluidics, cell culture, assay development, and multiphase systems. I recommend this manuscript for publication with the following considerations:

- Photos of side views should be included to show the device workflow and enable evaluation of contact angles referenced. A semi-side view (with some tilt) is given in Fig S8C, but it would be useful to have side views (taken directly from the side with no tilt) corresponding to the steps shown in Fig 1 and Fig S3 and the principles shown in Fig 2. These could be taken using a contact angle goniometer or a simpler homebuilt setup with a camera on a cheap commercially available mount. Taking side view images without artifacts of the aqueous fluid under FC40 may be challenging, but I think should be achievable put putting the sample in a large cuvette (or potentially a chamber slide – with straight walls, rather than a curved petri dish).

- Further context citing past work using aqueous biphasic systems for patterning should be included. Please see/cite Kaigala GV, Lovchik RD, Delamarche E. Microfluidics in the “open space” for performing localized chemistry on biological interfaces. *Angew Chem Int Ed Engl.* 5;51(45):11224-40. doi: 10.1002/anie.201201798 (see Figure 8 and references therein including subsequent work by Takayama’s group).

- It would be useful to explain that a key advantage of fluorinated oils in comparison to conventional oils for cell culture applications is that small molecules secreted by cells or drugs used to treat cells are less likely to partition into fluorinated oil than conventional hydrocarbon oils, PDMS channels, or aqueous biphasic systems. This could be added in the text near the section indicating that fluorinated oils are used widely in droplet-based microfluidics or later on when discussing the gas permeability of fluorinated oils in the cell culture section.

- Are any steps taken to prevent evaporation of the fluorinated oil? With the typical quantities of fluorinated oil used in this paper, how long can FF be maintained without having to top up the fluorinated oil? If a dish containing FF covered with fluorinated oil is placed in a cell culture incubator is there concern that the fluorinated oil vapors may be toxic to other cells in the incubator? The MSDS does provide some warnings about inhalation effects. It would be useful for the authors to share their observations on this practical point.

- I think it would be helpful to bring the quantification data from Fig S7 into the manuscript (in Fig 4). (Also typo: one caption indicates HEK cells and the other indicates BHK cells.)

- In the methods section the authors note, “Results obtained in Figure 5c were like those obtained with a commercial PDMS-based microfluidic system and PDMS-based devices fabricated in house (Oliveria et al., 2016).” Does this mean that the authors reproduced this work in house with PDMS

chips? If so it would be useful to include the data obtained in the Supporting Information so that readers could make the comparison. If this is not applicable, please clarify or remove the statement.

- Please include time stamps in all of the supplemental videos or indicate the factors by which the movies have been sped up.

- Comments on statistics (per reviewer instructions): Statistical tests are appropriate; the authors use ANOVA and post-hoc Tukey's test for multiple pairwise comparisons (Fig S7). Please indicate in the figure captions how many replicates (and are they within an experiment or from multiple independent experiments?) are represented by the error bars (Fig S7). For the representative images shown in Fig 4, please indicate in the caption how many replicates/independent experiments these images represent. Please include error bars in Fig 2B.

Ashleigh Theberge

Reviewer #2 (Remarks to the Author):

The manuscript, entitled "Microfluidics with fluid walls", describes a new way of generating the two-dimensional microfluidic circuits with a relatively simple setup, which requires no external driving force for the liquid movement. There present several advantages using such an approach, such as ultra-flexibility on circuit design, fully accessibility from the perpendicular dimension, and biocompatibility with the possibility of forming the constant flow and gradient, etc. The overall demonstration of the approach has been conducted in a relatively decent manner, together with the in-depth modelling analysis of the interplay of Laplace and hydrostatic pressure (Fig. S4), and proper application illustrations, including live-cell assays. Based on the originality and innovation, the reviewer would like to recommend this manuscript to be considered for publication, if all the following concerns were to be satisfactorily addressed in the revision.

The following are several questions / comments on the details of the manuscript:

1. Page 1, there should be a comma symbol after the name of Edmond J. Walsh.
2. Page 2, paragraph 2, last but one line, after 'challenges', there should be a colon instead of a semicolon; the last sentence, there should be some references after the 'trigger molecular aggregation at air-fluid interfaces.
3. Page 2, last paragraph, last line. The word 'footprint' appeared here for the first time. What's the definition of this word? Does it mean the area of contact between the drop and the substrate?
4. Page 3, first paragraph, second line, which says that the slight more medium can be added without the increase of the footprint in Fig. 1a_{iii}. However, from the figure, the footprint did increased between a_{ii} and a_{iii}. I wonder if this is a mistake in the figure, or it's not intended to be shown in the figure. If the latter reason is the case, then it's better to move the bracket '(Fig. 1a_{iii})' to the sentence before, in order to avoid misunderstanding.
5. Page 3, third paragraph. From the Fig. 1c, it seems that the drops on both ends of the conduit are formed together with the conduit with a single stroke. But from the Fig. 1d and Movie S1, it seems that the ends of a stroke could be relatively narrow/ small. Could the conduit be made without the forming of the drops on both ends? If so, then the FF circuits could be generated with higher efficiency as it takes different flow rate and move speed to generate the conduits and the drops. From Fig. 3e, I think it's possible, and it could usually be the trick to form the complicated microfluidics network. Please confirm this. Also, since it's possible to print the circuits under FC40,

is there a difference between overlaying the FC40 before the circuits are printed and after?

6. What's the thermal stability of FC40 as the overlay on the printed circuit? If there is instability on the surface among the aqueous media, substrate, and FC40, then will the FC40 go beneath the media due to density difference, i.e. buoyancy? Also, some liquid mixing operations will generate heat, and this might affect the shape of the conduit; if it's stable, then the system is very suitable to perform some temperature triggered experiments, such as PCR, or isothermal reactions.

7. Page 3, fourth paragraph, when generating the passive flow, is there a difference between polarized liquid and non-polarized liquid? Also, although the stability of the footprints are proved to be as long as several days, during the passive flow, will the widths of the conduits, or the size of the drops change?

8. Same paragraph, I noticed that in Fig. 2a_{ii} and in the Movie S3, the drops at the bottom conduit have the similar size, but eventually the dye diffused to the right hand side drop. My question is, if the dye is added to the larger side of the drop, will it also migrate to the smaller drop eventually due to the diffusion? If so, given enough time, will the dye be evenly diluted across the whole media, no matter how different the size of the drops are?

9. Page 4, last but three paragraph. It is said that Movie S5 illustrates gravity-driven splitting of a stream, but does it mean that the substrate has to be positioned vertically in order to let the dye flow toward the smaller sink drops, and if its positioned horizontally, then the flow is reversed, according to the theory in Figure 2a_{ii}? Since the direction of the channels are all roughly the same (facing down), if we combine the volume of the total 8 sink drops, it would be larger than the inlet drop. Can't we regard this microfluidic network as a single but very wide conduit with a large sink drop, so that the flow direction is still downward?

10. In the Movie S6, the middle portion of the fluidic media is not obvious in the mixing result. Is that very typical in the mixing? Or is it because of the color effect difference? How to confirm that the flow rate among these many inlets are controllable and repeatable?

11 Page 5, second paragraph. How long does the regrowth to allow TNF α to switch on GFP expression? According to the caption of Fig. 4c, I assume this incubation time is 24h. Similar to question/comment 8, during this incubation time, won't the concentration of the TNF α be constantly changing toward evenly distribution in the media due to diffusion? The effect of the drugs in each chamber should be related to the integration of the concentration during the whole incubation time (since it's changing), whereas the paper only measured the end-point concentration of the drug at 24h. Additionally, in Fig S7c, the concentration of the TNF α were calculated using the curve in a and b, which doesn't make much sense because fluorescence in a and b were generated by fluorescein, and the fluorescence in c was generated by GFP. Please explain these two issues, otherwise this paper suffers from logic flaws that are not eligible for publish.

12. Page 5, second paragraph, first sentence. Is there any reference to support the claim that the stable flows are required persisting for days?

13. Same paragraph, 9th line, that the footprint wouldn't be changed by injecting the red and blue dyes into the circuit. However, there should be a limit on the volumes of the dye injection before the footprint is altered. Have the authors tested such limitation?

14. Same paragraph, last 4th line, and the third paragraph, last but one line: when showing that the bubbles won't be a problem in this paper's approach, the authors only showed the bubbles introduced from the inlets; on the other hand, when talking about the bubble problems in PDMS-based devices, bubbles were travelling through the devices, detaching cells from surfaces and

unbalances flows and gradients. In my opinion, these two comparisons are not at the same level, as it's not that clear that the freestyle fluidics is a proper way of performing cell-related experiments, at least at the similar level as high as the relatively mature PDMS-based microfluidic chips.

15. Page 5, fourth paragraph, third line. According to the paper, the circuit was printed using DMEM + PBS, and washed through with excess TB. During the washing how much TB was used as 'excess'? How was this washing conducted? Was there a suction mechanism needed? Is it necessary to remove the DMEM + PBS in the sink? If not, then the drop size in the sink will become bigger and bigger. Will this affect the subsequent experiments?

16. Same paragraph, last sentence in the square brackets, which describes an alternative way of generating stable pinning lines. I'm curious how the surface could be modified to give stable pinning in TB, and why the authors prefer using DMEM + PBS to generate the pinning rather than modifying the surface.

17. Page 6, first paragraph. It is strongly suggested that the authors provide a figure comparing the results between the FF circuits and conventional PDMS-based devices, instead of merely providing a reference, to emphasize the similarity of the effectiveness and many other advantages of the FF circuits in cell migration experiments.

18. Page 6, second paragraph, which talked about the usage of a syringe pump to drive the pipetting pen. This paper keeps saying the cost-effectiveness of the system, especially compared with PDMS-based microfluidic chips. In fact, the utilization of a syringe pump is not cheap either, whereas PDMS-based chips could be driven by a simple pneumatic valve.

19. Same paragraph, last 9th line, where the fluid could be removed from selected drops, but this function is not mentioned much. How easy and effective this is? Will the drop be stable after the fluid removal? Keep in mind that the density of FC40 is larger than that of the aqueous media.

20. I think there should be more discussion needed before the summary could be even drawn. Such as the draw-back and limitation of the system, and issues like the easiness of building and operating such system in the lab, speed and throughput of the experiment conducting, etc.

21. Page 11, Figure 4a. It'll be a good idea to also put a scale bar in the left image. And, are these two images the same field of view?

22. Page 14, in the caption of the Figure S2 (ii), are 'remain sterile' and 'remain clear' the same meaning?

23. Page 15, The ABC in the figure S3 are upper-case, whereas the abc in caption text are lower-case.

24. Page 22, last paragraph. When building the system, is there any leveling procedure needed? It seems that the distance ($\sim 100 \mu\text{m}$) is very critical to the successful of printing the correct footprint or pinning.

25. Page 23, first paragraph. When pouring the FC40, is there any cautions needed? Will the flowing of the FC40 affect the shape of the aqueous circuit printed on the substrate? Although Movie S2 shows that the circuits can survive violent agitation of FC40 that is already overlaid, I'm afraid that the pouring is even more violent to local circuits.

26. Page 24, second paragraph, last line in the square bracket. The paper only proved that the system has the potential of being used in the screening of single drug, but when talking about screening two-drug combinations, I'm afraid a one-dimensional array of concentration is not

enough. Unless the authors could come up with a two-dimensional serial dilution design for two reagents, this sentence is advised to be taken away from the manuscript.

Reviewer #3 (Remarks to the Author):

Walsh et al. is a fascinating and fun read. This paper will attract a lot of attention and discussion.

The paper describes the writing of liquid lines and spots on a substrate wet by the written structures. After the writing, the structures are overlaid with an immiscible fluid. In this case, the structures were aqueous and deposited on glass or polystyrene and the immiscible fluid was biocompatible FC40. The authors characterized and modeled the forces generating flow between spots with a view to limited programmability. They also demonstrated that fluids that did not wet unmodified surfaces well could replace fluids that worked better, and the fluid with less compatible wetting properties would remain properly pinned on the surface—this demonstration addresses a major difficulty in using the system. I also really liked the discussion of how the system with fluid walls gets around problems with bubbles that plague microfluidics in general. The authors demonstrated bacterial and mammalian cell culture in the system and the ability of fabricated devices to be used for chemotaxis measurements. The figures and SI videos do an excellent job of supporting the conclusions of the authors. The English was succinct, yet clear, and organized in a very logical and easy-to-follow format.

My main criticism is that while the authors do a great job demonstrating and listing the advantages of this system, they really need to add a discussion on limitations:

*Clearly, the substrate, writing fluids and pinning fluids need to be carefully matched. What are the restrictions for doing this matching and the considerations/limitations that must be met?

*What are the size and volume limitations?

*What are the requirements for conditions of use (e.g. this might be a great system for adaptation to high throughput screening, but not at all practical for use by nontechnically trained individuals or in an unstable, dirty environment)?

The authors probably have a very good feel for what the system should not be used for by now—please share these insights with readers. Readers not familiar with the issues in the field may not appreciate these limitations and be disappointed that the approach does not work for them.

Reviewer #1 (Remarks to the Author):

- Photos of side views should be included to show the device workflow and enable evaluation of contact angles referenced. A semi-side view (with some tilt) is given in Fig S8C, but it would be useful to have side views (taken directly from the side with no tilt) corresponding to the steps shown in Fig 1 and Fig S3 and the principles shown in Fig 2. These could be taken using a contact angle goniometer or a simpler homebuilt setup with a camera on a cheap commercially available mount. Taking side view images without artifacts of the aqueous fluid under FC40 may be challenging, but I think should be achievable put putting the sample in a large cuvette (or potentially a chamber slide – with straight walls, rather than a curved petri dish).

We have now added a new side view (as Fig. S4a) using a rectangular well-plate as suggested, plus a new Movie taken from the side (as new Movie S4). The latter shows pipetting blue dye into a drop (much as in the cartoon in Fig. 1 so the drop increases in volume), and then pressure-driven flow out of the first drop (which then shrinks) through a conduit into a second drop (which enlarges) – as in Fig. 2.

- Further context citing past work using aqueous biphasic systems for patterning should be included. Please see/cite Kaigala GV, Lovchik RD, Delamarche E. *Microfluidics in the "open space" for performing localized chemistry on biological interfaces. Angew Chem Int Ed Engl.* 5;51(45):11224-40. doi: 10.1002/anie.201201798 (see Figure 8 and references therein including subsequent work by Takayama's group).

The suggested review and one primary reference added.

- It would be useful to explain that a key advantage of fluorinated oils in comparison to conventional oils for cell culture applications is that small molecules secreted by cells or drugs used to treat cells are less likely to partition into fluorinated oil than conventional hydrocarbon oils, PDMS channels, or aqueous biphasic systems. This could be added in the text near the section indicating that fluorinated oils are used widely in droplet-based microfluidics or later on when discussing the gas permeability of fluorinated oils in the cell culture section.

This has been changed as suggested, with the insertion into the section headed 'Human cells grow normally in FF drops and circuits' of 'These circuits can be constructed using biocompatible liquids (i.e., tissue-culture media), substrates (i.e., polystyrene Petri dishes/glass slides), and overlay (FC40 is bio-inert, permeable to the vital gases – O₂ and CO₂, and small molecules secreted by cells and/or added drugs are less likely to partition into this fluorinated oil compared to a conventional hydrocarbon oil, PDMS channel, or aqueous biphasic system).'

- Are any steps taken to prevent evaporation of the fluorinated oil? With the typical quantities of fluorinated oil used in this paper, how long can FF be maintained without having to top up the fluorinated oil? If a dish containing FF covered with fluorinated oil is placed in a cell culture incubator is there concern that the fluorinated oil vapors may be toxic to other cells in the incubator? The MSDS does provide some warnings about inhalation effects. It would be useful for the authors to share their observations on this practical point.

Added as suggested, and 'Methods' now begins: 'FC40 was purchased from Acota. It is bio-inert, and not found in regulatory lists of dangerous organic chemicals (<http://www.acota.co.uk/assets/data-centre/msds/3m/3mfc40msds.pdf>). If circuits are to be kept for days, extra FC40 should be added when needed, and we give the following data as a rough guide

to the amounts needed. The vapour pressure of FC40 is 432 Pa at 25°C and so evaporates relatively slowly compared to water (vapour pressure of 3170 Pa), and we find experimentally that the rate of evaporation of FC40 at 25°C from a 60-mm Petri dish is 90 µl/day with the lid on, and 1.55 ml/day with the lid off (assessed by weighing the dish). Evaporated FC40 also had no untoward effects on any of many different cell types grown conventionally in the same incubator at the same time over a period of 2 years.'

- I think it would be helpful to bring the quantification data from Fig S7 into the manuscript (in Fig 4). (Also typo: one caption indicates HEK cells and the other indicates BHK cells.)

Combining Figs 4 and S7 is easily done – but more than doubles the length of what is already a long legend (we are at your word limit). As we think the original version of Fig. 4 tells the story simply, and that the interested reader can go to Fig. S7 for the detail, we prefer to leave things as they are. However, we will change it if you think it necessary and the word limit can be extended.

- In the methods section the authors note, “Results obtained in Figure 5c were like those obtained with a commercial PDMS-based microfluidic system and PDMS-based devices fabricated in house (Oliveria et al., 2016).” Does this mean that the authors reproduced this work in house with PDMS chips? If so it would be useful to include the data obtained in the Supporting Information so that readers could make the comparison. If this is not applicable, please clarify or remove the statement.

No, we were referring to an independent experiment published in PNAS. The published paper described the use of both commercially-made and home-made devices; for the sake of clarity, we refer now only to results obtained with the former. We also reproduce some of the previously-published data in a different form (new panel Fig. S8e) to allow the reader to compare results obtained in an FF conduit and PDMS channel.

- Please include time stamps in all of the supplemental videos or indicate the factors by which the movies have been sped up.

Where time-lapse movies do not contain time stamps, the legend accompanying each Movie states how many times the movie was speeded up from real time.

- Comments on statistics (per reviewer instructions): Statistical tests are appropriate; the authors use ANOVA and post-hoc Tukey’s test for multiple pairwise comparisons (Fig S7). Please indicate in the figure captions how many replicates (and are they within an experiment or from multiple independent experiments?) are represented by the error bars (Fig S7). For the representative images shown in Fig 4, please indicate in the caption how many replicates/independent experiments these images represent. Please include error bars in Fig 2B.

Re Fig S7: The data is from 3 circuits using cells plated on the same day. This is now stated.

Re Fig. 4: The images in Fig. 4C are from chambers from one of the 3 circuits. This is now stated.

Re Fig. 2B: Figure 2B provides data from a single experiment. To illustrate the requested reproducibility, we add the new data shown in the graph in Fig. S4A.

Reviewer #2 (Remarks to the Author):

1. Page 1, there should be a comma symbol after the name of Edmond J. Walsh.

Corrected.

2. Page 2, paragraph 2, last but one line, after ‘challenges’, there should be a colon instead of a semicolon; the last sentence, there should be some references after the ‘trigger molecular aggregation at air-fluid interfaces.’

Corrected, and reference added.

3. Page 2, last paragraph, last line. The word 'footprint' appeared here for the first time. What's the definition of this word? Does it mean the area of contact between the drop and the substrate?

This definition now included.

4. Page 3, first paragraph, second line, which says that the slight more medium can be added without the increase of the footprint in Fig. 1a_{iii}. However, from the figure, the footprint did increase between a_{ii} and a_{iii}. I wonder if this is a mistake in the figure, or it's not intended to be shown in the figure. If the latter reason is the case, then it's better to move the bracket '(Fig. 1a_{iii})' to the sentence before, in order to avoid misunderstanding.

Now clarified, and the new sentence reads: 'After printing the drop, slightly more medium can be added without the footprint increasing in area, with the exact amount being determined by the advancing contact angle. But once this angle is reached, footprint area increases (Fig. 1a_{iii}).'

5. Page 3, third paragraph. From the Fig. 1c, it seems that the drops on both ends of the conduit are formed together with the conduit with a single stroke. But from the Fig. 1d and Movie S1, it seems that the ends of a stroke could be relatively narrow/ small. Could the conduit be made without the forming of the drops on both ends? If so, then the FF circuits could be generated with higher efficiency as it takes different flow rate and move speed to generate the conduits and the drops. From Fig. 3e, I think it's possible, and it could usually be the trick to form the complicated microfluidics network. Please confirm this. Also, since it's possible to print the circuits under FC40, is there a difference between overlaying the FC40 before the circuits are printed and after?

Re: single stroke/drops on both ends: This is correct; FF circuits can be formed either way. Using continuous strokes and then adding drops by pipette at the ends of lines is quicker. However, from the videos one can see the time taken is increased by order tens of seconds between both methods. Therefore, the time to make a circuit in either way is still relatively rapid. Forming drops while making a circuit has the key advantage that the footprint area is well defined, and then the amount of fluid to be added to provide a known pressure can be calculated using the theory in SI (i.e., different volumes added to the same wetted footprint give different radii of curvature and hence different pressures). Although easily achieved, creating strokes and then adding a drop manually has the disadvantage that the shape of the drop then depends on the operator (and so is less repeatable compared to using the printer). This is now clarified by inclusion in Methods: 'Complex circuits can either be printed using continuous flow (as in Fig. 2a and Movie S1), or by stopping/starting the pump (and retracting the pen from the surface) as individual features in a circuit (and even parts of a feature) are made.'

Re adding FC40 first or last: For all experiments in this paper except Movie S3 and Fig. S3c, circuits were made in air and then overlaid; therefore, we have not evaluated printing under FC40 in much detail as it is not the focus of this paper (but the main difference is the printing tip needs to be closer to the substrate assuming other conditions remain unchanged – and this point has now been added to Methods).

6. What's the thermal stability of FC40 as the overlay on the printed circuit? If there is instability on the surface among the aqueous media, substrate, and FC40, then will the FC40 go beneath the media due to density difference, i.e. buoyancy? Also, some liquid mixing operations will generate heat, and this might affect the shape of the conduit; if it's stable, then the system is very suitable to perform some temperature triggered experiments, such as PCR, or isothermal reactions.

This is an important question, especially for the future. We have demonstrated stability at the temperatures described here (i.e., we have not detected any change in pinning line over days of operation up to 37°C). We also have never found that the FC40 goes beneath the media. However, we have not considered how higher temperatures (PCR temperatures) and/or internal heat generation affect stability so we cannot comment with certainty (but we have no reason to believe

that as long as the substrate can handle the temperature, and the boiling point of the fluids is not reached that the method will not work). So, possibly a topic for a future paper. [Text not changed, as this is a comment.]

7. Page 3, fourth paragraph, when generating the passive flow, is there a difference between polarized liquid and non-polarized liquid? Also, although the stability of the footprints are proved to be as long as several days, during the passive flow, will the widths of the conduits, or the size of the drops change?

Re polarized liquids: We assume this concerns non-polar and polar liquids. If so, polarity can have an effect through interfacial properties, but the general theory remains the same – so we have not modified the text. [If a concern is that polarity might change local temperatures due to local reactions at the interface (point 6), this might change interfacial tensions, but note that the ratio of surface area to volume is large so heat is rapidly dissipated from a drop.]

Re changes in size of drops and conduit widths. No, footprints do not change, so maximum widths of drops and conduits do not change. However, the height, and hence shape, above footprints (of both drops and conduits) does change. We have now added (i) a new Movie (i.e., Movie S4) illustrating this.

8. Same paragraph, I noticed that in Fig. 2a_{ii} and in the Movie S3, the drops at the bottom conduit have the similar size, but eventually the dye diffused to the right hand side drop. My question is, if the dye is added to the larger side of the drop, will it also migrate to the smaller drop eventually due to the diffusion? If so, given enough time, will the dye be evenly diluted across the whole media, no matter how different the size of the drops are?

There are several points here: First, drops in the bottom conduit do not have the same sizes (see Legend; they may look similar at first glance since the length scale that you see is proportional to the cube root of volume – so a drop of half the volume in this view would only have a diameter increase as seen in this movie of ~ 1.26). Therefore, a difference in Laplace pressure drives flow (resulting in advection, not diffusion, between drops). Second, diffusion may equalize concentrations eventually, but it is not effective over the lengths and times discussed here. We address this point on diffusion in detail and have modified the text as described in the response to point 11 below. Using the details of point 11, the diffusion time for the circuit with drops separated by 30 mm (as in Movie S3) and for typical proteins would be around 50 days.

9. Page 4, last but three paragraph. It is said that Movie S5 illustrates gravity-driven splitting of a stream, but does it mean that the substrate has to be positioned vertically in order to let the dye flow toward the smaller sink drops, and if its positioned horizontally, then the flow is reversed, according to the theory in Figure 2a_{ii}? Since the direction of the channels are all roughly the same (facing down), if we combine the volume of the total 8 sink drops, it would be larger than the inlet drop. Can't we regard this microfluidic network as a single but very wide conduit with a large sink drop, so that the flow direction is still downward?

Re 'does it mean that the substrate has to be positioned vertically': Yes. [Text unchanged.]

Re 'if it is positioned horizontally, then the flow is reversed, according to the theory in Figure 2a_{ii}?': Yes. [Text unchanged.]

Re 'Can't we regard this microfluidic network as a single but very wide conduit with a large sink drop, so that the flow direction is still downward?': Yes. In this case, gravity dominates and hence the size/number or shape of the sink drops is somewhat arbitrary in terms of flow (as suggested). [Text unchanged.]

10. In the Movie S6, the middle portion of the fluidic media is not obvious in the mixing result. Is that very typical in the mixing? Or is it because of the color effect difference? How to confirm that the flow rate among these many inlets are controllable and repeatable?

Re mixing of the middle portion. The objective here is to demonstrate flow focussing (and mixing through diffusion in the direction perpendicular to the flow which causes a change in the colour of the central stream). The clear fluid reduces in width, and hence the diffusion distance for the colored dyes is less. Fig. S6a and b show flows in the same device, but here the red and clear fluids in the source drops have been reversed (by pipetting out one fluid by hand and replacing it with the other) and then the central stream is red so that this becomes more obvious. [Text unchanged.]

Re reproducibility: If we have the same geometry, fluids, and substrates, we have the same interfacial tensions, and so physics provides reproducibility. Again, Figure S6 provides an example of reproducibility (where the same circuit is reused); we also add a new Figure (i.e., new Fig. S4b) that shows reproducible flow rates between chambers from independent experiments.

11 Page 5, second paragraph. How long does the regrowth to allow TNF α to switch on GFP expression? According to the caption of Fig. 4c, I assume this incubation time is 24h. Similar to question/comment 8, during this incubation time, won't the concentration of the TNF α be constantly changing toward evenly distribution in the media due to diffusion? The effect of the drugs in each chamber should be related to the integration of the concentration during the whole incubation time (since it's changing), whereas the paper only measured the end-point concentration of the drug at 24h. Additionally, in Fig S7c, the concentration of the TNF α were calculated using the curve in a and b, which doesn't make much sense because fluorescence in a and b were generated by fluorescein, and the fluorescence in c was generated by GFP. Please explain these two issues, otherwise this paper suffers from logic flaws that are not eligible for publish.

Re time to switch on GFP expression: It was 24 h (as stated). [Text unchanged.]

Re changing concentrations: We agree we should have determined if (1) molecules of TNF α transfer between test chambers, and (2) the concentration of TNF alpha changes. We have now performed the necessary calculations and included them in Methods as follows: 'The initial transport of TNF α from source drop to cell chambers is by advection; once advection ceases, diffusion becomes the transport mechanism. Therefore, two questions arise: how far does TNF α diffuse, and by how much does diffusion change concentrations during the experiment? To provide theoretical answers, we use the diffusion coefficient (D) of a typical protein – the green fluorescent protein – in water (i.e., $\sim 1 \times 10^{-6}$ cm²/s; Swaminathan *et al.*, 1997). [A value for TNF α is 3×10^{-7} cm²/s, but we use the larger value for the green fluorescent protein both to be more conservative and to provide a more general result.] The diffusion distance, x , of this protein over 24 h (t) is 4.15 mm (calculated using the established relationship $x = \sqrt{2Dt}$). Therefore, the green fluorescent protein would diffuse ~ 4.15 mm from a chamber into a conduit in one day, and take ~ 11 days to diffuse from one chamber to another. Consequently, there will be no diffusional transfer of proteins between cell chambers during our experiment. However, the concentration in a chamber may change over time due to diffusion into a conduit, and this change can be calculated using theory provided in **Supplementary Data**. The volume of a conduit in **Figure 4** that connects cell chambers (for a diffusion length of 4.15 mm) is 0.03 μ l, compared to a chamber volume of 3.6 μ l. If the average concentration over this diffusion length is half that in the chamber, then only 0.4% of a protein will be lost from chamber to conduit in 24 h.'

Re determining the concentration of TNF α : This is related to the calibration procedure, and we think this may be a misunderstanding. GFP is expressed in response to TNF α , and fluorescein is used to characterize the concentration of TNF α in a drop. The use of fluorescein is valid here since advection generates the dilution series (not diffusion), and molecules in the media (either TNF α or fluorescein) will have the same end concentration level when advection ceases. Therefore, fluorescein is a suitable surrogate for TNF α (or any other molecule). As stated in the legend to Fig. S7a, fluorescein is used as a surrogate for TNF α , so the text is unchanged.

12. Page 5, second paragraph, first sentence. Is there any reference to support the claim that the stable flows are required persisting for days?

Reference now cited.

13. Same paragraph, 9th line, that the footprint wouldn't be changed by injecting the red and blue dyes into the circuit. However, there should be a limit on the volumes of the dye injection before the footprint is altered. Have the authors tested such limitation?

Footprints remain unchanged (as stated) throughout the 9 h of the experiment. We now restate this, adding that longer times can be accommodated 'by removing fluid from the sink by pipette, or using a flat sink of larger diameter'. [If fluid were not removed, and sink size increased, it would start to alter the length of the circuit.] We have also run this circuit at $300 \mu\text{l h}^{-1}$ without altering pinning lines, and we now state this in Methods.

14. Same paragraph, last 4th line, and the third paragraph, last but one line: when showing that the bubbles won't be a problem in this paper's approach, the authors only showed the bubbles introduced from the inlets; on the other hand, when talking about the bubble problems in PDMS-based devices, bubbles were travelling through the devices, detaching cells from surfaces and unbalances flows and gradients. In my opinion, these two comparisons are not at the same level, as it's not that clear that the freestyle fluidics is a proper way of performing cell-related experiments, at least at the similar level as high as the relatively mature PDMS-based microfluidic chips.

We understand the reviewer's comments. The reason for this comparison is as follows. In PDMS devices, a bubble enters a channel either through an inlet or forms on an internal wall (both are common), and it must then flow through the microfluidic network towards the exit. In doing this, experimental failure is often the result – which is well known to users of PDMS-based devices. However, the key point here is that when bubbles enter an FF circuit they do NOT flow through the conduit (they float to the surface, as illustrated in Movie S10) and hence potential issues with them do not occur. We have never noticed a bubble forming in an FF conduit during flow in the many experiments over the past two years. Therefore, the current text is addressing the general problem of bubbles in microfluidics by comparing PDMS and FF devices.

Re maturity: Again we agree with this comment; however, the results in this paper are a start towards demonstrating the one can at least do some things using FF that one can also do in PDMS-based devices, and that FF may provide a powerful alternative to PDMS-based microfluidics in the longer term (especially when using living cells). [Text unchanged.]

15. Page 5, fourth paragraph, third line. According to the paper, the circuit was printed using DMEM + PBS, and washed through with excess TB. During the washing how much TB was used as 'excess'? How was this washing conducted? Was there a suction mechanism needed? Is it necessary to remove the DMEM + PBS in the sink? If not, then the drop size in the sink will become bigger and bigger. Will this affect the subsequent experiments?

Comment: it was 'DMEM + FBS' and not 'DMEM + PBS'.

Re 'how much TB was used in excess' and 'How was this washing conducted?': details are now added to Methods (in summary, it was pumped in through needles and any amount could be used).

Re 'Was there a suction mechanism needed? Is it necessary to remove the fluid in the sink?': We have added to Methods: 'Next, most fluid pumped into the sink up to this stage was removed manually from the sink by pipetting.' [This means that after we wash out and before we run the chemotaxis experiment, we empty the sink using a pipette; therefore, there was no effect of washing on the subsequent experiment.]

16. Same paragraph, last sentence in the square brackets, which describes an alternative way of generating stable pinning lines. I'm curious how the surface could be modified to give stable pinning

in TB, and why the authors prefer using DMEM + PBS to generate the pinning rather than modifying the surface.

The key objective here was to use unmodified glass (so we could reproduce previous results, and – more fundamentally – allow those working with glass to continue to work on the surface they understand). However, there are many ways of modifying glass (or other) surfaces chemically (or otherwise) to alter their surface properties, and – as stated – we didn't explore these for the reason given, and we also found this method very effective. [Text unchanged.]

17. Page 6, first paragraph. It is strongly suggested that the authors provide a figure comparing the results between the FF circuits and conventional PDMS-based devices, instead of merely providing a reference, to emphasize the similarity of the effectiveness and many other advantages of the FF circuits in cell migration experiments.

We now provide side-by-side panels showing results obtained using FF and a conventional PDMS-based device (some previously-published data is reproduced in a different form in the new panel in Fig. S8e to make results comparable). [The chemotactic effect is simply indicated by the red zones being of greater area than the blue ones, indicating cell movement is biased in one direction.]

18. Page 6, second paragraph, which talked about the usage of a syringe pump to drive the pipetting pen. This paper keeps saying the cost-effectiveness of the system, especially compared with PDMS-based microfluidic chips. In fact, the utilization of a syringe pump is not cheap either, whereas PDMS-based chips could be driven by a simple pneumatic valve.

It is correct that syringe pumps can be expensive like the Harvard Elite/Ultra we use, but several companies supply others for a few hundred pounds. However, FF can be adapted to be used with most fluid movers (e.g., a peristaltic pump) as there is no particular need for a syringe pump. Here, we wanted well-defined control to create long-term steady gradients, and that is why we used a high-quality syringe pump with Hamilton air-tight syringes.

Re pneumatic valve: While you can operate PDMS chips with a pneumatic valve, the valve must be connected to some fluid mover/pressure source. We also see no obvious reason why the same pumping mechanism could not be used here, by connecting such a valve to the printing tip. [Text unchanged.]

19. Same paragraph, last 9th line, where the fluid could be removed from selected drops, but this function is not mentioned much. How easy and effective this is? Will the drop be stable after the fluid removal? Keep in mind that the density of FC40 is larger than that of the aqueous media.

This is trivial, you simply pipette as you would normally – except now through FC40 instead of air. For example, we add fluid this way for Figures 2a_{ii}, the circuits of Figure 3 (when uncovered or covered with FC40), the circuit of Figure 4 (mammalian cells), Figure 5 (bacteria), and Figure S6 (adding and removing fluids from drops to reverse the dyes in the source drops). To bring this out we have added to the end of the third paragraph in Results: 'aqueous liquids are simply pipetted into (and removed from) drops through FC40 instead of air (**Fig. 1b**; **Methods** gives further detail).'

20. I think there should be more discussion needed before the summary could be even drawn. Such as the draw-back and limitation of the system, and issues like the easiness of building and operating such system in the lab, speed and throughput of the experiment conducting, etc.

We now add a paragraph on drawbacks/limitations of the system (see response to referee 3 which provides the details).

Re speed/throughput/ease of use: We hope that the new discussion of these limitations, plus the videos showing various circuits being made and fluid added to them provides the reader with a good feel for what time/expertise and additional materials that might be required.

21. Page 11, Figure 4a. It'll be a good idea to also put a scale bar in the left image. And, are these two images the same field of view?

The two frames are from a movie showing a single field of view, so the scale bar is the same for both. [Text unchanged.]

22. Page 14, in the caption of the Figure S2 (ii), are 'remain sterile' and 'remain clear' the same meaning?

Clarified by inserting 'visibly', so we now say: 'remain visibly clear'.

23. Page 15, The ABC in the figure S3 are upper-case, whereas the abc in caption text are lower-case. Corrected.

24. Page 22, last paragraph. When building the system, is there any leveling procedure needed? It seems that the distance (~100 μm) is very critical to the successful of printing the correct footprint or pinning.

We have added to Methods: 'Beds of the system and printer were levelled using miniature bulls-eye spirit levels. The needle tip was brought to within ~100 μm of the surface of a horizontal 60-mm dish by first lowering the tip until it touched the surface and then raising it 100 μm .' [Comment: the value of 100 μm is what we used, but it can be any value as long as the liquid touches the surface as the pen moves (so that pinning lines are formed) – for example, we have used between 40-200 μm .

25. Page 23, first paragraph. When pouring the FC40, is there any cautions needed? Will the flowing of the FC40 affect the shape of the aqueous circuit printed on the substrate? Although Movie S2 shows that the circuits can survive violent agitation of FC40 that is already overlaid, I'm afraid that the pouring is even more violent to local circuits.

We have never noticed this to be an issue. However, it is only prudent to pour the FC40 into the dish away from the circuit, and we now state: 'Prudence dictates that FC40 is poured next to (and not directly on to) a circuit, but experience indicates that pinning lines are usually strong enough to remain unchanged even when FC40 is poured directly onto circuits (Figure S3C shows part of a circuit that was created while FC40 flowed over it).'

26. Page 24, second paragraph, last line in the square bracket. The paper only proved that the system has the potential of being used in the screening of single drug, but when talking about screening two-drug combinations, I'm afraid a one-dimensional array of concentration is not enough. Unless the authors could come up with a two-dimensional serial dilution design for two reagents, this sentence is advised to be taken away from the manuscript.

Sentence removed.

Reviewer #3 (Remarks to the Author):

My main criticism is that while the authors do a great job demonstrating and listing the advantages of this system, they really need to add a discussion on limitations:

**Clearly, the substrate, writing fluids and pinning fluids need to be carefully matched. What are the restrictions for doing this matching and the considerations/limitations that must be met?*

**What are the size and volume limitations?*

**What are the requirements for conditions of use (e.g. this might be a great system for adaptation to high throughput screening, but not at all practical for use by nontechnically trained individuals or in an unstable, dirty environment)?*

The authors probably have a very good feel for what the system should not be used for by now— please share these insights with readers. Readers not familiar with the issues in the field may not appreciate these limitations and be disappointed that the approach does not work for them.

We have dealt with these points largely through the addition of a new paragraph in the Discussion: 'As with any method, FF has limitations. (i) Fluids and surfaces must be matched to ensure pinning lines are stable. Fortunately, suitable combinations can be screened rapidly by placing a drop of fluid on a substrate, and then removing most of the volume. If the pinning line does not retract, the combination may be used; if it retracts, a fluid known to allow fabrication of FF circuits can be replaced by one desired for operation (as in **Fig. 5c**). (ii) Many existing PDMS circuits cannot be replicated exactly, so new designs to achieve existing functions must be developed. For example, the fluid walls in FF circuits are curved and not straight, FF conduits have lengths and widths larger than many PDMS channels (we typically use printing tips with diameters of hundreds of microns), source drops have maximal diameters of ~5 mm (giving an upper volume of ~20 μ l for a contact angle of 70°), and pressures tolerated are lower because of the limitations imposed by interfacial forces (**Supplementary data**). (iii) Fluids used in cell biology are often transparent, so FF circuits are difficult to see; therefore, we often print the circuit plan on paper, place it under the dish, and then use the plan as a guide to aid manually pipetting into circuits. (iv) Circuits should usually be horizontal during operation (achieved using a bulls-eye spirit level).'

I do hope you find these corrections satisfactory.

One final point: I would like to include the Ed Walsh as a corresponding author (he is the engineer, I am the biologist), but I cannot find any way of doing this electronically (but I have it on the manuscript).

With best wishes,

Peter

REVIEWERS' COMMENTS:

Reviewer #1 (Remarks to the Author):

The authors have addressed my concerns, and I recommend this manuscript for publication.
-Ashleigh Theberge

Reviewer #2 (Remarks to the Author):

The authors carefully replied to all of the questions I brought up, and answered almost all my concerns to the system they built. I will only add several more comments/ questions as follows, before the paper is accepted for publication:

For the issue of adding the FC40 before the media printing (point 5), since for all experiments in this paper except Movie S3 and Fig. S3c, circuits were made in air and then overlaid, I suggest that authors mention this detail (that most experiments were carried out before the FC40 overlay) in the main article. Also, in Methods, the fact that the tip needs to be closer to the substrate in the case of FC40 overlay first than after was not mentioned.

On page 26, starting line 662, since most experiments were carried out by the pouring of the FC40 after printing the conduit, during the actual experiments, was FC40 poured next to a circuit due to prudence, or directly onto circuits since it doesn't matter? What's the height and speed of pouring? Do you have any data presenting that pouring directly onto the circuit won't affect the pinning lines?

About the evaporation. The paper only measured the evaporation of the FC40 in room temperature (page 25, line 607), but there is no description of evaporation of cell media (the aqueous phase that is printed). Since the FC40 is poured after the printing, from the first stroke till the last stroke, the resulting volume and concentration within the circuit array or droplet array would change due to different evaporation time. To compensate such change, necessary actions should be taken. Otherwise, the authors need to prove that the effect of evaporation of media is negligible (for example, the printing is fast enough).

About the thermal stability (point 6). Although it makes sense to predict that stability is likely there all the time as long as the boiling point is not reached, the inner circulation of liquid will increase due to the temperature rise. Therefore, I suggest the authors to mention in the paper that the conclusion that the drop (page3, line 87) and FF circuits (page 3, line 105) overlaid with FC40 is stable for days was achieved under room temperature, or no higher than 37-degree C, to avoid confusion.

In response to point 9: i) if the substrate has to be positioned vertically, then will the FC40 be flowing downwards also? Will this be a problem since the container is open to the air? ii) My original question was: if we can regard the combination of all the outlet drops as a large single drop, then due to the Laplace theory, the flow direction should be reverse. And yet it still flows down, so does it mean that the Laplace force is negligible compared with gravitational force?

Reviewer #3 (Remarks to the Author):

My comments have been well addressed.

We have addressed the comments of referee 2 as described below.

Reviewer #2 (Remarks to the Author):

For the issue of adding the FC40 before the media printing (point 5), since for all experiments in this paper except Movie S3 and Fig. S3c, circuits were made in air and then overlaid, I suggest that authors mention this detail (that most experiments were carried out before the FC40 overlay) in the main article. Also, in Methods, the fact that the tip needs to be closer to the substrate in the case of FC40 overlay first than after was not mentioned.

We now include a statement as suggested in Methods.

On page 26, starting line 662, since most experiments were carried out by the pouring of the FC40 after printing the conduit, during the actual experiments, was FC40 poured next to a circuit due to prudence, or directly onto circuits since it doesn't matter? What's the height and speed of pouring? Do you have any data presenting that pouring directly onto the circuit won't affect the pinning lines? We now include a new Movie (Supplementary Movie 12) showing that adding FC40 isn't a problem. Since it was never an issue, we did not do detailed analysis on speeds/distances etc – and we think that the new movie plus original Supplementary Movie 2 confirms this.

About the evaporation. The paper only measured the evaporation of the FC40 in room temperature (page 25, line 607), but there is no description of evaporation of cell media (the aqueous phase that is printed). Since the FC40 is poured after the printing, from the first stroke till the last stroke, the resulting volume and concentration within the circuit array or droplet array would change due to different evaporation time. To compensate such change, necessary actions should be taken. Otherwise, the authors need to prove that the effect of evaporation of media is negligible (for example, the printing is fast enough).

The reviewer is indeed correct, there will be some evaporation from start of printing to the end. This is because the quantity of fluid used to make a circuit is typically at least an order of magnitude less than what is added during operation, so any effects of evaporation are reduced. However, for flow circuits (e.g., our chemotaxis experiment), this is irrelevant as flow is continuous and hence the media concentration will be whatever is pumped in. For passive circuits or drops, the quantity of fluid used to make a circuit is typically at least an order of magnitude less than what is added during operation, so any effects of evaporation are reduced. In practice, circuits are made in seconds and the various cell experiments show that as far as the various types of cells we used are concerned, any evaporation doesn't affect their growth and behaviour (response to drugs, chemotaxis). If one believed evaporation was an issue in an experiment, one could simply wash out a circuit with a fluid of choice, before adding cells.

About the thermal stability (point 6). Although it makes sense to predict that stability is likely there all the time as long as the boiling point is not reached, the inner circulation of liquid will increase due to the temperature rise. Therefore, I suggest the authors to mention in the paper that the conclusion that the drop (page 3, line 87) and FF circuits (page 3, line 105) overlaid with FC40 is stable for days was achieved under room temperature, or no higher than 37-degree C, to avoid confusion.

Included as suggested.

In response to point 9: i) if the substrate has to be positioned vertically, then will the FC40 be flowing downwards also? Will this be a problem since the container is open to the air? ii) My original question was: if we can regard the combination of all the outlet drops as a large single drop, then

due to the Laplace theory, the flow direction should be reverse. And yet it still flows down, so does it mean that the Laplace force is negligible compared with gravitational force?

Yes, the FC40 will flow downwards, and over time will evaporate (and then the circuit will also evaporate). This was shown as something that could be done with the method – and (as stated) intention was not to address all the benefits/caveats of every design shown but to give the reader a flavour of the possibilities.

Best,

Peter